# The Use of Nuclear Magnetic Resonance Spectroscopy (NMR) to Characterize Bitumen Used in the Road Pavements Industry: A Review

**DOI:** 10.3390/molecules29174038

**Published:** 2024-08-26

**Authors:** Dilshad Shaikhah, Cesare Oliviero Rossi, Giuseppina De Luca, Ruggero Angelico, Pietro Calandra, Paolino Caputo

**Affiliations:** 1Institute of Functional Surfaces, School of Mechanical Engineering, University of Leeds, Woodhouse Lane, Leeds LS2 9JT, UK; d.m.shaikhah@leeds.ac.uk; 2Scientific Research Center, Soran University, Erbil 44008, Kurdistan Region, Iraq; 3Department of Chemistry and Chemical Technologies, University of Calabria, UdR INSTM della Calabria, Via P. Bucci, Cubo 14/D, 87036 Rende, CS, Italy; giuseppina.deluca@unical.it (G.D.L.); paolino.caputo@unical.it (P.C.); 4Department of Agricultural, Environmental and Food Sciences (DIAAA), University of Molise, Via De Sanctis, 86100 Campobasso, CB, Italy; 5CNR-ISMN, National Research Council, Institute of Nanostructured Materials, Via Salaria km 29.300, 00015 Monterotondo, RM, Italy; pietro.calandra@cnr.it

**Keywords:** bitumen, Nuclear Magnetic Resonance Spectroscopy (NMR), Physical Chemical Characterization, Colloids

## Abstract

Bitumen, a vital component in road pavement construction, exhibits complex chemo-mechanical properties that necessitate thorough characterization for enhanced understanding and potential modifications. Nuclear Magnetic Resonance (NMR) spectroscopy emerges as a valuable technique for probing the structural and compositional features of bitumen. This review presents an in-depth exploration of the role of NMR spectroscopy in bitumen characterization, highlighting its diverse applications in determining bitumen content, group composition, molecular dynamics, and interaction with additives. Various NMR techniques, including free induction decay (FID), Carr–Purcell–Meilboom–Gill (CPMG), and Pulsed Field Gradient Stimulated Echo (PFGSE), are discussed in the context of their utility in bitumen analysis. Case studies, challenges, and limitations associated with NMR-based bitumen characterization are critically evaluated, offering insights into potential future research directions. Overall, this review provides a comprehensive overview of the current state-of-the-art in NMR-based bitumen characterization and identifies avenues for further advancement in the field.

## 1. Introduction

### 1.1. Background on Bitumen Used in Road Pavements

The terms bitumen and asphalt are often used interchangeably, but they are not the same. Bitumen is a naturally occurring material that is made up of hydrocarbons, while asphalt is a man-made product derived from bitumen. Both materials are widely used in road construction and other building projects, but there are some key differences between them. Most roads are constructed using asphalt, which is a mixture of bitumen and other materials such as sand, gravel, and stone. The bitumen acts as a binding agent to hold the other components together and provide strength to the road surface. It also helps to protect the road from water damage and wear caused by traffic.

Bitumen is the basis of most roads and provides it with strength, durability, and protection from water damage. Different combinations of sand, gravel, stone, and other materials can create an even more durable surface that is better able to withstand the wear caused by traffic.

The construction of a bitumen road typically involves several layers of materials, including a sub-base layer, a base layer, and a wearing course layer. The sub-base layer pro-vides support for the road, the base layer provides additional strength, and the wearing course layer is the visible surface that vehicles drive on.

Bitumen is a fundamental binder in road pavements and constructions, possessing unique chemo-mechanical properties that make its use critical in its diverse applications. The present work is to be interpreted as part of a bigger plan to improve the knowledge about how bitumen characteristics and properties can be understood. For this purpose, a brief presentation of bitumen’s chemical composition and microstructural organization is here given.

Bitumen is basically a visco-elastic fluid, with a semi-solid aspect at room temperature. Obviously, viscosity is always temperature-dependent, so, at higher temperatures, typically above 60 °C, a marked softening takes place, with the bitumen becoming a viscous Newtonian liquid. Generally, it is derived from crude oil distillation, so its chemical nature is mostly apolar, non-volatile, adhesive, and waterproofing. Due to its origin, the chemical composition and consequently the physico-chemical detailed characteristics strongly depend on crude oil sources undergoing the topping distillation process [1,2]. All these peculiarities have made bitumen a pivotal component throughout the world for roofing and industrial applications but, above all, due to its role as a binder of inorganic aggregates, in asphalt concrete production for road paving. Asphalts are, in fact, biphasic systems, with macro-meter-sized inorganic aggregates making a predominant phase (about 93–96% w/w, inorganic particle size from microns to millimeters, following well-defined granulometric curves for specific characteristics [3] hold together by small amounts of binding bitumen (c.a. 5 wt %) which constitutes the second phase. Although bitumen constitutes only a minor part of the asphalt, it plays the most important role in determining the final material properties, since it gives global consistency.

From a chemical point of view, bitumen is a complex mixture of Saturates, Aromatics, Resins, and Asphaltenes. Usually, these fractions can be determined by the so-called S.A.R.A. determination (Saturates, Aromatics, Resins, and Asphaltenes) [4] and are typically used as a fingerprint of the material, giving a representative composition of the molecules present in the bitumen, divided into classes.

Slight changes in the bitumen composition (like in the spontaneous evaporation process of the most volatile components, or in the presence of specific additives) can dramatically affect the overall properties of an asphalt. Even the usability of asphalt can be traced back to the properties of its bitumen.

This is because the intermolecular self-assembly, giving the final structural pattern, follows a complex behavior, with a delicate equilibrium among all the interactions among all these components, which is at the basis of the conditions for asphaltic flow.

The final structure is usually described with analogies of colloidal systems: the dispersed phase is made up of micelles of high polarity and molecular mass made by asphaltenes stacks, behaving as solid particles. These colloidal aggregates are dispersed in the dispersing phase, an oily apolar environment of lower molecular weight known as maltene, granting fluidity [5].

The size, abundance, and interconnection of asphaltene clusters are of utmost importance in dictating the bituminous material properties. Causes such as the aggression of chemicals normally present in the environment, or aging can trigger the oxidation of some of the organic components and therefore their increase in polar functional groups, with an ultimate clustering/immobilization of polar molecules giving bitumen embrittlement and susceptibility to cracks. So, generally, the abundance and the aggregation pattern of asphaltenic fraction can be related to the rigidity of the bitumen.

The dispersing medium, i.e., the apolar maltene phase, in turn, is composed of saturated paraffins, aromatic oils, and resins, as schematically depicted in Figure 1. The proposed model of bitumen is called the colloidal model: the asphaltene, in the form of polar nano-aggregates, is stabilized by resins that behave as the dispersing agents for asphaltene molecules combining them with aromatics and saturates; however, asphaltenes have the tendency to self-assemble into hierarchical structures of different length-scales [6].

The structure therefore resembles that of reversed micelles in water-in-oil microemulsions (polar micelles dispersed in apolar matrix, such as in bitumen). So, following the analogy with the micellar model, and borrowing the information known for such systems, the stabilization of the polar domains is of pivotal importance.

This phenomenon is governed mainly by the stabilizing agents present in the bitumen, mainly resins. Their peculiar characteristic can be drawn back to amphiphilic properties, i.e., those properties possessed by molecules with both polar and apolar parts within their molecular architecture. So, at their polar side, they can bind the polar clusters of asphaltenes, and at the apolar one, they can bind the maltene. In this way, they act as a bridge between polar and apolar domains in the bituminous system. With this mechanism, they actually hold up the overall structure of the system. In the absence of such molecules, the domains with opposite polarity would be unstable toward segregation/separation/sedimentation. Without going deep into details, it is important to point out here that they have a complex role in the overall self-assembly and dynamics, determining the size and structure of the dispersed aggregates. This is, however, a principle of general validity being observed in a wide range of different materials dispersed in the organic matrix: organic-based particles [8,9], inorganic complexes [10], and even nanoparticles [11]. Interestingly, it has been recently highlighted that, in addition to polar and apolar interactions, further specific interactions act between amphiphiles themselves with consequent peculiar mutual interactions and self-assembly processes of the extended molecular network, dictating the final overall aggregation pattern and the (usually slowed-down) dynamics. For all these reasons, we feel that bituminous materials are very sensitive to their composition and that some effects, especially when additives are added with the aim of improving their properties, are even unpredictable.

As ever, in this framework, the chemistry of bitumen is the key aspect to define its physical properties, so there are mutual relationships between intermolecular interactions, aggregates structures, and their dynamic properties, the rheology. In addition, the behavior of bitumen is dependent not only on its structure but also on the maltene’s glass transition temperature and the effective asphaltene content [12]. The main problem encountered when dealing with bitumen is that the molecules involved are not uniquely defined, so bitumen is classified according to the molecular weight and polarity of its components. Ought to the presence of millions of such constituents, the chemical analysis of bitumen is usually performed based on the molecular structure class, not by studying the molecular species individually [13].

For this reason, our belief is that any technique differentiating the types of molecules or fractions probing differences in their contribution to the overall system property would be a highly effective form of analysis, but we have some doubts that this would be feasible or of practical use. Instead, easy-to-apply methods of analysis giving immediate information, as well as more sophisticated techniques aiming at exploring deeper details of the overall structure/behavior of the bituminous material or specific aspects are, in our opinion, better strategies of approach. For these, some introductive information is given in the next paragraph.

### 1.2. Importance of Characterizing Bitumen

Characterizing a bitumen is a hard task since its characteristics are not trivial at all: it is a highly viscous, organic-based, viscoelastic microheterogeneous fluid made of nano-meter-sized polar aggregates of asphaltenes dispersed in a more apolar continuous phase of saturated paraffins, aromatic oils, and resins called maltene [12,14]. The most immediate approach is therefore to characterize the relative amounts of these fractions. The so-called S.A.R.A. determination (Saturates, Aromatics, Resins, and Asphaltenes) [4] aims at this by exploiting the different solubility of each of these fractions to selected solvents: Saturated components of the sample are developed in *n*-heptane solvent, aromatics in a 4:1 mixture of toluene and *n*-heptane, and resin in a mixture generally made of 95/5% dichloromethane/methanol, whereas the asphaltene fraction is left on the lower end of the rods. The following are worth noting:Asphaltenes are not defined by their molecular structure, but rather on the basis of the procedure required to extract them from heavy oils [15];Bitumen is a not well-defined mixture of constituents, so different methods of analysis exist where solvents are added to bitumen to determine its chemical properties.

For this reason, other methods are unavoidably present. Zenke [16], for example, offered an improved method distinguishing light-, middle-, and high-solubility asphaltenes. In this framework, however, we believe that the detailed chemical speciation of the various molecules is of secondary importance with respect to their overall assembly if an effective study needs to be carried out. In this optics, bituminous materials are regarded in terms of their rheological characteristics. Empirical approaches are always followed to determine the performances within a chosen temperature range [17,18] for convenient use [19]. Penetration index, softening point, ductility, and viscosity are quick and easy properties to determine, and the low cost certainly encourages approaches based on their use. However, a rational structure/properties correlation based on molecular interpretation is always a hard task in such complex systems such as bitumens. In these cases, the direct correlation between molecular composition, molecular assembly, and final overall properties cannot be as feasible as in simple systems due to the arising of emerging properties [20]. On the other side, more sophisticated investigative tools are present to probe the structure. Generally, such methods are carried out by Atomic Force Microscopy (AFM) [21], Confocal Laser Scanning Microscopy [22], optical microscopy [23], and fluorescence microscopy [24], but all these techniques can probe either the micro-scale, ruling out the possibility to gain molecule-based information at the nano-scale, or the surface, ruling out the possibility to bulk information. Gaining bulky structural information at the nano-scale suffers from limited attempts, and the results remain quite hypothetical [25]. Even the “colloidal structure” is just empirically derived by the contents of aromatics, resins, asphaltenes, and saturates [26]. This is probably due, in our opinion, to the always-urgent need to improve performances for applicative purposes, which is often answered by the addition of additives, so that basic research, highlighting the specific intermolecular interactions and the molecular organization at the base of the observed behavior, has sometimes been overlooked. Therefore, there is still a lack of information on how many additives affect the supra-molecular structure and the distribution of aggregates within the bituminous colloidal network and how this can reflect on the overall material properties. This makes the relationship between molecular interactions and the final material structure/properties (which is ultimately the final objective of physical chemistry) still quite vague.

Scattering experiments, and in particular X-ray scattering ones, would be advisable to probe the structure from the Å to the meso-scale. As a matter of fact, remarkable attempts have been made since the 1960s [27], but the bitumen complex organization has hindered the development of such structural studies in detail. In fact, it has been recently recognized that bitumen is characterized by intermolecular associations at different length scales: asphaltene molecules are aggregated to form stacks, similar to those of polycyclic aromatic rings, of about 18 Å, which in turn are organized at higher levels of complexity in anisotropic aggregates of about 200 Å × 28 Å, which, again, are assembled to form micrometer-size elongated aggregates characterized by the so-called “bee-structure” [28]. However, asphaltenes do not form a continuous network [29] since they and their aggregates are stabilized by polar resins, i.e., those molecules with amphiphilic behavior present in the material and, finally, all these aggregations are dispersed in a paraffin-like apolar matrix. This complex organization makes the material a complex system, for which emerging properties arising from the opportune organization of the molecules can also be expected.

The potentialities of X-ray scattering make such an approach a method of election [27] for the investigation of structural aspects, which of course need to be compared to the properties measured by other techniques.

In this framework, an alternative method also possessing the advantage of probing dynamical aspects is constituted by Nuclear Magnetic Resonance (NMR), which is the focus of this contribution. For this, some introductory information showing its role in bitumen characterization is given in the next paragraph.

### 1.3. Role of Nuclear Magnetic Resonance (NMR) Spectroscopy in Bitumen Characterization

Nuclear Magnetic Resonance (NMR) spectroscopy emerges as a pivotal tool in bitumen research and the petroleum industry, offering versatile applications [30]. It facilitates the determination of bitumen content and group composition in rock samples without the need for extraction, utilizing a combination of free induction decay (FID) and modified Carr–Purcell–Meilboom–Gill (CPMG) measurements through low-frequency (LF) NMR relaxation techniques [31]. NMR spectroscopy, in combination with other analytical methods [32], has proven useful for performing the compositional characterization of substances belonging to the two main fractions of which bitumen is composed, i.e., resins and asphaltenes [33]. NMR spectroscopy is crucial for studying the structural formulas and spatial and electronic structures of organic compounds in petroleum products, providing detailed insights into the composition of complex mixtures such as bitumen. Indeed, NMR techniques such as time domain (TD) NMR have been proposed as a new methodology to quantify asphaltene precipitation in crude oil, offering advantages in terms of time and cost savings compared to traditional methods such as ASTM D6560 [34,35,36]. One- and two-dimensional NMR spectroscopy is a powerful tool in characterizing the molecular structures and aggregation processes of the most polar components in bitumen, known as asphaltenes, despite their complexity [37]. The results of in-depth NMR analysis of bitumen structure and chemistry can be successfully correlated to complementary analytical techniques such as FT-IR, fluorescence spectroscopy, and elemental analysis, and provide a powerful multi-spectroscopy method for the elucidation of structural changes of bitumen upon aging processes [38,39].

In the context of unconventional reservoirs, where shale rock characterization presents complexities due to the presence of producible hydrocarbon fluids and semi-solid organic phases such as bitumen, NMR plays a crucial role [40]. By employing combined FID and CPMG measurements, along with advanced inversion methods, NMR enables accurate differentiation of the bitumen signal from added water in shale samples, facilitating precise analysis [41]. Moreover, NMR provides insights into water infiltration in porous bitumen–salt matrices, elucidating leaching behavior and material evolution over time [42]. Additionally, NMR spectroscopy allows for the assessment and quantification of the aliphatic hydrogen components relative to the aromatic fraction of bitumen [43]. Furthermore, NMR-based examination of molecular dynamics through techniques such as NMR Pulsed Field Gradient Stimulated Echo (PFGSE) and relaxometry (CPMG spin-echo pulse sequence) offers valuable insights into the effects of polymers on the bitumen system and the aging phenomenon of bitumen over time [44]. Parameters such as self-diffusion coefficients (D) and transverse relaxation times (T_2_) play a pivotal role in capturing these dynamic variables [45,46].

In essence, the diverse applications of NMR spectroscopy underscore its significance in facilitating the characterization of the complexities of bitumen, thereby driving advancements in road pavement industry practices and materials engineering. By exploiting the ability of NMR to provide valuable information on the molecular architecture, composition, and aging processes of bitumen, researchers can gain a deeper understanding of bitumen’s properties and develop increasingly high-performance and sustainable pavement materials. Thus, this review aims to revisit and compare various NMR techniques employed in bitumen characterization. Through the presentation of case studies, challenges, and limitations encountered in NMR-based bitumen analysis, this review provides a comprehensive understanding of the current landscape. Furthermore, it offers insights into potential future directions for utilizing NMR in bitumen characterization, thereby paving the way for advancements in this field.

## 2. Nuclear Magnetic Resonance (NMR) Spectroscopy

### 2.1. Basic Principles of NMR Spectroscopy

Nuclear Magnetic Resonance (NMR) exploits the magnetic properties of certain atomic nuclei, characterized by their nuclear spin quantum number (I) and gyromagnetic ratio (γ). Nuclei are NMR active if I > 0 and they have a significant γ value [47]. For instance, 1H has a spin of 1/2 and a γ of 267.522 × 10^6^ rad s^−1^ T^−1^, with nearly 100% natural abundance. NMR involves inducing nuclear spin precession using a magnetic field, where the precession frequency, or Larmor frequency (ω), depends on the magnetic field strength (B₀), as described by the following equation [48]:ω = γB₀(1)

To generate an NMR signal, radio-frequency (RF) pulses (B₁) are applied perpendicular to B₀, exciting nuclear spins from their equilibrium state and enabling observation of their precession. The decay of this signal, known as relaxation, reveals molecular dynamics and is useful for studying adsorption, confinement, and pore characteristics in porous media.

### 2.2. Relaxometry Analysis

NMR relaxation consists of longitudinal (T_1_) and transverse (T_2_) relaxation. Longitudinal relaxation (T_1_), or spin-lattice relaxation, describes the return of the magnetic moment to thermal equilibrium, as modeled by
(2)$$\frac{M_{z}\left(\tau_{1}\right)}{M_{0}}=1-2exp\left(-\frac{\tau_{1}}{T_{1}}\right)$$
where M_z_/M_0_ is the longitudinal magnetization recovery, τ₁ is the delay time, and T_1_ is the relaxation time constant [49]. T_1_ is determined using an inversion recovery (IR) pulse sequence with a 180° RF pulse followed by a 90° RF pulse.

Transverse relaxation (T_2_), or spin-spin relaxation, involves the dephasing of magnetic moments due to local magnetic field inhomogeneities [50]. The CPMG pulse sequence refocuses this dephasing, allowing T_2_ measurement:(3)$$\frac{M_{xy}\left(nt_{e}\right)}{M_{0}}=exp\left(-\frac{nt_{e}}{T_{2}}\right)$$
where *M*_xy_/*M*_0_ represents the transverse magnetization decay, *t_e_* is the echo time (in seconds), *n* is the number of echoes, and *T*_2_ is the transverse relaxation time constant (in seconds).

To obtain relaxation time distributions, the Fredholm integral equation is used:(4)$$\frac{M\left(t\right)}{M_{0}}=\int F\left(T_{1,2}\right)K_{1,2}d\left(logT_{1,2}\right)+\varepsilon \left(t\right)$$
where *F*(*T*_1_,_2_) is the distribution of relaxation time constants, *K*_1,2_ is the kernel function describing the expected form of NMR relaxation data, and *ϵ*(*t*) represents experimental noise.

This inversion is stabilized with Tikhonov regularization, minimizing as follows:∣∣*M*−*KF*∣∣ + *α*∣∣*F*∣∣^2^(5)

The level of smoothness, governed by the smoothing parameter α, is optimized using a robust algorithm known as generalized cross-validation (GCV) [51].

Usually, the T_2_ relaxation time varies all over the sample because of the sample heterogeneity or surface relaxation differences. A method for determination of the real T_2_ of the sample is to apply the Carr−Purcell technique (CP). In a complex system, a multiexponential attenuation of the CP envelope should be observed. Hence, if inside the sample a continuous distribution of relaxation times exists, the amplitude of the nth echo in the echo train is given by
(6)$$A_{n}=A_{0}\;\int_{0}^{\infty }P(T_{2})e^{\frac{-2n\tau }{T_{2}}}\;dT_{2}$$
where P(T2) is the T_2_ relaxation time probability density. Equation (6) suggests that the analysis of the experimental data using an inverse Laplace transform (ILT) might provide the relaxation time probability function. The ILT is a well-known mathematical tool, and it can give the answers required in several fields (spectroscopic techniques, digital electronics, and so on) where it needs to face the inverse problem of estimating the desired function from the noisy measurements of experimental data. Anyway, for convenience, we shortly recall the definition of the problem. Let f(t) be a function defined for t ≥ 0; the function F(s) introduced by means of the expression
$${F\left(s\right)=\left\{Ff\left(t\right)\right\}=\int_{0}^{\infty }e}^{-st}f\left(t\right)dt$$
which is the real Laplace transform of f(t). The inverse process, indicated by the notation f(t) = L^−1^[F(s)], is termed the inverse Laplace transform (ILT).

The spin−spin relaxation time (T_2_) of a material can be a measure of molecular mobility under certain conditions. The relaxation process is, in fact, more efficient when the material is more rigid; this corresponds to shorter relaxation times. As matter of the fact, higher mobility corresponds to longer T_2_ times. Hence, the proton T_2_ time distribution can be used to evidence structural changes when rigid and soft parts coexist in the materials.

### 2.3. Diffusometry

NMR diffusometry distinguishes itself from relaxation measurements through the use of magnetic field gradients [52]. Pulsed-field gradient (PFG) diffusion measurements encode and decode nuclear spin phase shifts using magnetic field gradients [53]. The phase shift, determined by the gyromagnetic ratio (γ), gradient strength (g), and spin position difference, signifies diffusion.

The decay of PFG NMR signal due to diffusion is modeled by
*S*_0_/*S*_(*g*)_ = exp(−*Dγ*^2^*g*^2^Δ)(7)
where D is the self-diffusion coefficient and Δ is the diffusion period. The PFG pulse sequence, such as the pulsed field gradient spin echo (PFGSE) sequence, is employed to measure diffusion [54]. In rocks, PGSTE sequences are often preferred due to their longer diffusion times.

Additionally, the 13-interval bipolar alternating pulsed gradient stimulated echo (APGSTE) pulse sequence is utilized to improve diffusion measurements in heterogeneous systems [55]. This sequence effectively minimizes coherent gradient echoes and undesirable spins, enhancing signal acquisition and accuracy.

## 3. Types of NMR Employed in Bitumen Characterization

### 3.1. One Dimensional NMR

One-dimensional ^1^H and ^13^C NMR spectroscopy plays a crucial role in studying the microstructure of crude oil due to its ability to provide detailed information on the composition and molecular architecture of complex materials such as bitumen. These techniques are essential for assessing the aliphatic and aromatic components of bitumen, allowing for a comprehensive understanding of its chemical composition and structural properties. Additionally, NMR spectroscopy has been utilized to investigate the molecular structure and aggregation processes of highly polar components such as asphaltenes, which are key constituents of bitumen [36,56]. Below, some examples drawn from the literature are briefly summarized.

Mono-dimensional proton NMR (1D ^1^H NMR). The ^1^H NMR spectrum of crude oil with 9% asphaltene (sample 1) using a Bruker Avance II 300 Hz (Bruker, Karlsruhe, Germany) is shown in Figure 1a. Perdeuterated toluene was added for field stability. Peaks at 2.2 and 7.3 ppm are from protonated toluene. Without toluene, weak, broad peaks of other aromatic protons are observed. Aliphatic protons are seen between 0.5 and 1.9 ppm. The inset of Figure 1a zooms in on the aliphatic region, showing a CH_2_/CH_3_ intensity ratio of 2.07, suggesting octane. Doublets due to J couplings of 9 and 12 Hz for CH_3_ and CH_2_ groups are noted. Figure 1b compares ^1^H NMR spectra of crude oil with and without asphaltene, showing a ratio of ~0.097 for the small peak around 1.12–1.17 ppm, aligning with the 9% asphaltene content. The main difference is the narrowing of aliphatic peaks and the decrease in the small peak around 1.17 ppm, likely due to the reduced presence of asphaltene aggregates.

1D ^13^C NMR Spectrum. The ^13^C NMR spectrum (5–60 ppm for aliphatic, 100–160 ppm for aromatic carbons) was discerned on a Bruker Avance 600 MHz NMR spectrometer. Figure 1c shows the spectrum of crude oil with 9% asphaltene (sample 1), and Figure 1d shows the spectrum without asphaltene (sample 2). A septuplet around 20 ppm corresponds to the coupling to the methyl group of perdeuterated toluene. Comparing the ^13^C intensities relative to the septuplet shows an 11.4% increase in sample 1 compared to sample 2, consistent with the 9% asphaltene found by chemical analysis.

### 3.2. Two-Dimensional NMR

Two-dimensional (2D) NMR measurements are pivotal for concurrently determining relaxation time constants T_1_ and T_2_. This is typically achieved through inversion recovery combined with CPMG measurements. The resulting NMR relaxation signal is represented as a linear combination of two exponentials, delineating T_1_ and T_2_, as outlined by Venkataramanan et al. (2002) [57]. The acquired 2D-NMR data adhere to the principles of the first-kind Fredholm integral equation, presenting significant computational challenges due to an exponential increase in the number of values. Singular value decomposition (SVD) and Kernel separability are utilized for data compression and simplification, rendering the resulting minimization problem manageable.

A valuable application involves correlating transverse relaxation time and diffusion coefficients (T_2_-D), facilitating robust applications such as distinguishing between signals from oil, gas, and water. This correlation is achieved through the integration of PFGSE and CPMG experiments [58,59]. Figure 2a shows the (T_2_-T_2_) relaxation exchange REXSY measurement pulse sequence. The thin and thick vertical lines represent 90° and 180° radio frequency pulses, respectively. The gray rectangle indicates the presence of a magnetic field spoiler gradient. The sequence consists of three-time intervals: *t*_A_, *t*_B_, and *t*_S_. The encoding intervals *t*_A_ and *t*_B_ contain CPMG echo trains with m and n number of echoes, respectively. The time between successive echoes in each train is 2τ. Data (echo intensities) are only acquired during the second CPMG train. The middle interval, *t*_S_, is the storage time over which exchange is observed.

Furthermore, Figure 2b illustrates the raw data from a T_2_-T_2_ exchange experiment, with a storage time of *t*_S_ = 750 ms. The S/N ratio is better than 104 in this data. Figure 2c is the T_2_-T_2_ spectrum (obtained from sample 2, *t*_S_ = 2 s), showing the manually defined peak integral boundaries (solid rectangles). The error in the integration was determined by increasing or decreasing the boundaries by one relaxation time division on the x- and y-axes (dotted rectangles). This method provided an estimate of the maximum and minimum peak intensities and was particularly applicable to the integration of merged peaks.

T_2_-T_2_ exchange spectra from water-saturated borosilicate and soda lime glass spheres are notable in sample 2, with storage delays of Figure 3. (a) t_S_ = 50 ms, (b) 2 s, and (c) 4 s. The exchange peaks develop symmetrically on either side of the diagonal T_2_^A^ = T_2_^B^, becoming more clearly resolved as the storage delay increases. The contour intervals are the same in all of these plots.

The processing and inversion of all 2D NMR data can be accomplished using the method outlined in Figure 4, employing the Fredholm integral equation of the first kind [60]. This review examines mathematical methods for processing NMR relaxation or diffusion data to obtain multi-dimensional distributions. Key stages shown in Figure 4 include data format and pre-processing, kernel matrix generation, singular value decomposition (SVD), truncated singular value decomposition (TSVD), and three inversion methods: NNLS, maximum entropy, and Tikhonov regularization with optimization techniques. Ultimately, 2D outputs from each method are compared to each other using simulated data.

### 3.3. FFC-DNP Fast Field Cycling Dynamic Nuclear Polarization

The rotational correlation time of vanadyl complexes is slow enough to result in Dynamic Nuclear Polarization (DNP) enhancement through the solid effect while yielding liquid-state NMR spectra of protons, which have a much faster rotational correlation time scale. See, for example, [61,62] (Figure 5).

### 3.4. DOSY

Diffusion-ordered 2D-NMR spectroscopy (DOSY) is the high-resolution version of the PFGSE sequence of Stejskal and Tanner [63]. It is based on the construction of a two-dimensional spectrum, where the first dimension is a typical NMR spectrum illustrating the chemical shift of molecular species and the second one represents their diffusion coefficients. Since this technique was developed, it has proven to be very powerful in enabling chemical resolution and identification of mobility of the different constituents in complex molecular mixtures, thus providing valuable information on size and aggregation state [64].

As shown in Figure 6a–c, ^1^H DOSY spectra display the molecular self-diffusion behaviors of heavy crude oil and its fractions (i.e., Resin and SAoil) in 2D contour maps. The signal intensity of each diffusion map arises mainly from the contributions of H_β_ and H_γ_ as well as H_α_ protons, and the peak signals can be assigned to the protons of H_β_ or/and H_γ_. The additional diffusion projection curve on the left is the projection outline of the top signal intensity of the entire diffusion map on the diffusion dimension. More information can be found in [64].

In each ^1^H DOSY spectrum, the residual chloroform in deuterated chloroform solvent is clearly separated from the diffusion maps of the petroleum components. It is identified by the single signal at 7.25 ppm, and its diffusion coefficient (*Dchl*) is variable. In fact, the trace amount of non-deuterated chloroform can act as probe molecules. The *Dchl* increases from 2.01 × 10^−9^ m^2^ s^−1^ in heavy crude solution via 5.40 × 10^−9^ m^2^ s^−1^ in Resin solution to 1.13 × 10^−8^ m^2^ s^−1^ in SAoil solution. This result indicates that the inhibition effect on chloroform diffusion is sequentially weakened, reflecting that the overall attractive interactions between solute–solute molecules and between solute–solvent molecules in the solutions of Heavy Crude, Resin and SAoil are weakened in this order.

Figure 6d displays D_r_ values of SAoil > Resin > Heavy Crude (D_r,HC-2_ > D_r,HC-1_ > D_r,HC-3_). This order correlates with molecular weights: SAoil (Mw 261, Mn 193) < Heavy Crude (Mw 1501, Mn 297) < Resin (Mw 2448, Mn 513). SAoil shows the largest D_r_ due to its small MWs and narrow distribution. Resin’s smaller D_r_ contradicts its wide MW distribution, explained by the aggregation-to-convergence effect. Rapid molecular exchange in resin solutions, with weak intermolecular interactions, leads to uniform sizes resembling monodisperse systems, common in aggregate-containing systems such as micelles.

Durand et al. reported the first ^1^H DOSY spectrum of asphaltenes in toluene and found that polycyclic aromatic hydrocarbons were highly substituted and connected to long alkyl chains, a structure compatible with the continental model of asphaltenes [65]. No DOSY spectrum of asphaltenes exists in literature prior to Figure 7, which presents the first ^1^H DOSY spectrum of asphaltenes. In this spectrum, aliphatic signals dominate, while aromatic protons are undetected due to a low signal-to-noise ratio compared to toluene. Solvent signals can be isolated from asphaltene signals, with significantly different diffusion coefficients: around 2.4 × 10^−10^ m^2^ s^−1^ for asphaltenes and 20.1 × 10^−10^ m^2^ s^−1^ for toluene. Residual n-heptane signals can be extracted from solute signals. Structural information, including CH, CH_2_, and CH_3_ connected to polycyclic aromatic hydrocarbons, can be inferred. CH_2_ signals dominate, with CH_3_ signals from long alkyl chains. Molecular weight estimation yields approximately 3450 g·mol^−1^, higher than recent publications and likely underestimated. An average hydrodynamic radius of about 15 Å is estimated using the Stokes–Einstein equation.

In a subsequent work, the same authors tested the concentration dependence of the diffusion coefficients by analyzing ^1^H DOSY spectra and diffusion profiles of asphaltene to identify the separation of two main classes of aggregates of asphaltenes above a given concentration [66]. The ^1^H DOSY spectrum of a Buzurgan asphaltene was analyzed at two different concentrations, 0.1 and 10 wt % in toluene-d_8_. Initially, the toluene signal can be separated from the asphaltenes signals. Residual heptane signals were detected at very low concentrations, especially at the highest asphaltene concentration, due to the NF T60-115 extraction method. A notable molecular weight dispersion was observed in the asphaltene signals, particularly at 10 wt %, indicating high polydispersity at higher concentrations. While theoretically all components could be isolated by mobility, the complexity of asphaltenes makes this difficult. However, two distinct classes of aggregates were detected above 3 wt %.

The DOSY spectrum showed a separation between these two classes of asphaltene aggregates, a finding not previously reported. The diffusion dimension projection of DOSY spectra for aliphatic peaks (0–5 ppm) of asphaltenes at 0.1, 3, and 10 wt % confirms this (Figure 8a). Intensities are in arbitrary units and not normalized.

For the 0.1 wt % sample, the smallest peak was amplified tenfold compared to others. At this concentration, only one peak was seen. At 3 wt %, the distribution peak showed two shoulders for nanoaggregates (Figure 8, right) and macroaggregates (Figure 8, left). At 10 wt %, two distinct peaks appeared. The broad peak at 3 wt % split into two at 10 wt %, indicating a wide distribution of diffusion coefficients, suggesting various aggregates. Figure 6 and Figure 7 show two distinct zones of petroleum species besides the solvent signal. These figures highlight a key advantage of ^1^H DOSY: isolating two asphaltene classes overlapped in the ^1^H spectrum without assumptions on sample composition, crucial as the asphaltene composition is not fully known.

Figure 8b shows the relative diffusivity (D_sample_/D_solvent_) of Buzurgan asphaltenes by solute concentration. Samples in toluene-d_8_ were analyzed at 20 °C across 0.01–15 wt %. Average diffusion coefficients were from aliphatic peaks (0.7–5 ppm) for asphaltenes and aromatic peaks for the solvent (toluene) in the proton spectrum.

Both solvent and sample diffusion coefficients depend significantly on asphaltene concentration. Figure 8b displays relative diffusivity, using the solvent as an internal reference to eliminate viscosity and temperature dependence.

Relative diffusivities remain constant up to 0.25 wt %, indicating a dilute state where solute molecules are isolated and primarily interact with the solvent. In this range, solute–solvent interactions dominate, yielding limiting D values: D_∞,tol_ ∼ (19.57 ± 0.05) × 10^−10^ m^2^/s for toluene and D_∞,asp_ ∼ (2.4 ± 0.1) × 10^−10^ m^2^/s for Buzurgan asphaltenes. The D_∞,tol_ value aligns well with the pure toluene-d_8_ diffusion coefficient.

Beyond 0.25 wt %, increasing concentration introduces obstruction effects, significantly altering solvent diffusion properties. The solute’s D then varies nonlinearly due to intermolecular interactions, including solvent–solute and solute–solute attractions. D continuously decreases past 0.25 wt %, marking a regime shift from dilute (0–0.25 wt %) to semidilute (beyond 0.25 wt %) concentrations. This demonstrates that intermolecular interactions are highly concentration-dependent.

Analogous asphaltene dispersions in toluene solutions were investigated by Lisitza et al. by using the ^1^H DOSY NMR technique to determine the molecular sizes of both monomers and nanoaggregates and were also able to estimate the average number of molecules within an aggregate [67]. ^1^H-DOSY experiments using stimulated echoes, longitudinal eddy current delay, and bipolar pulsed field gradients, were exploited by Korb et al. to identify two populations of translational D, the slowest of which due to a small proportion of light hydrocarbons interacting with the asphaltene nanoaggregates and the fastest one coming to a larger proportion of hydrocarbons moving in between the asphaltene macroaggregates [36].

DOSY results for sample 1 at two temperatures are shown in Figure 9a,b and for sample 2 at room temperature in Figure 9c, displayed on the same scale and intensity. Projections of the ^1^H spectrum and distribution of D are shown. DOSY peaks were observed only in the ^1^H aliphatic range, supporting the continental model of asphaltenes with highly substituted aromatic cores. In asphaltene solutions, two hydrocarbon populations were identified with D for CH_2_ and CH_3_ groups separated by factors of 2.64 and 3.0, respectively (Figure 9a). At 294 K, two well-separated distributions of intensities 35% and 65% were centered on 0.65 and 1.9 × 10^−10^ m^2^/s. At 313 K, these shifted to 1.5 and 3.1 × 10^−10^ m^2^/s, indicating activated dynamic processes without local disentanglement or crystallization.

In the absence of asphaltene, the separation factor reduced to 1.25 for CH_2_ and 1.5 for CH_3_ peaks, indicating a single hydrocarbon population with the slower D distribution merging into the faster one (Figure 9c). Analysis of projected D distributions (Figure 9a inset) showed the best fit with monomodal or bimodal log-normal distributions. For the low diffusion peak (35% hydrocarbons), a unique pdf mode at 0.63 × 10^−10^ m^2^/s was found. For the higher diffusion peak, a bimodal pdf with modes at 1.77 × 10^−10^ m^2^/s (6.5% hydrocarbons) and 1.85 × 10^−10^ m2/s (58.5% hydrocarbons) was observed.

With a diffusion delay of 20 ms, translational D is sufficiently fast to traverse macroaggregates, consistent with the observed diffusion length (5 μm) and average macroaggregate spacing (50 nm). These projected D distributions align well with chain length distributions from GC and GPC, suggesting two hydrocarbon populations: slower diffusing small hydrocarbons interacting with asphaltene nanoaggregates and faster diffusing hydrocarbons moving between macroaggregates.

The physicochemical behavior of asphaltenes obtained from three sources of crude oils extracted from Brazilian wells was investigated by da Silva Oliveira et al. with the DOSY NMR technique in a wide range of concentrations in toluene solvent [68]. The authors observed three different aggregates for the asphaltenes in toluene termed, respectively, nanoaggregates, microaggregates, and macroaggregates, according to an increasing order of size.

Figure 10 shows DOSY spectra of three asphaltenes at 8% concentration in deuterated toluene. Figure 10a illustrates sample asph_A, displaying signals related to toluene’s chemical shifts with D = 16 × 10^−10^ m^2^/s. This spectrum reveals that nanoaggregates and microaggregates do not separate well, as seen in Figure 2. At this concentration, nanoaggregates and microaggregates form a single signal, while macroaggregates, with a smaller D form another signal.

In the DOSY spectra of asph_B and asph_C (Figure 10b,c), the separation between nano, micro, and macroaggregates is clearer. D of toluene varies for each sample, confirming it depends on the concentration, shape, composition, and temperature of the mixture. Figure 6, Figure 7 and Figure 8 show correlations between the relative D (D_asp_/D_tol_) and concentration for the samples.

Figure 11a shows nanoaggregate and microaggregate formation for asph_A in diluted solutions, with macroaggregates forming at concentrations above 1%. In Figure 11b, macroaggregate formation begins at 0.1%, indicating a higher affinity between aggregates, likely due to the continental type of this asphaltene. Asphaltene B, from lighter oil with less asphaltene content, shows larger aggregate formation at lower concentrations, confirming asphaltene precipitation is related to oil type and composition.

Figure 11c indicates that for asphaltene C (asph_C), macroaggregates form from a concentration of 1.0% in toluene-d_8_, similar to asph_A. Both asph_A and asph_C come from heavier oils than oil B. Asphaltenes from oil B do not dissolve above 8% in toluene, showing high aggregation stability, explained by π–π stacking interactions of the continental type aromatic system.

A key finding is an intermediate aggregation state between nanoaggregates and macroaggregates. If intermolecular interactions do not favor the archipelago and continental models, the intermediate state, or microaggregate, represents aggregation between larger and smaller chains.

The high-field 2D DOSY NMR method, combined with low-frequency 2D NMR techniques such as D-T_2_, T_1_-T_2,_ and nuclear magnetic relaxation dispersion (NMRD), was applied directly on bituminous samples by Vorapalawut et al. [69]. They were characterized by different asphaltene concentrations, formulated by diluting a crude oil with maltene, in order to preserve the structure of the asphaltene aggregates initially present in the native bitumen. The authors confirmed the restriction of a two-dimensional translational diffusion of short hydrocarbon chains (maltene) within asphaltene nanoaggregates.

Figure 12a,b show 2D 1H DOSY spectra for native crude oil, 4.55% wt asphaltene crude oil, and maltene. The horizontal axis represents chemical shifts in ppm, and the vertical axis represents D in units of 10^−12^ m^2^/s. DOSY peaks are observed only in the 1H aliphatic range (0.5–1.9 ppm), with no aromatic peaks likely due to short relaxation times.

The bimodal DOSY peaks in Figure 12 resemble those from the D−T_2_ measurement at 23 MHz for native crude oil (9 wt % asphaltene) at long T_2_, with two D peaks at 2 × 10^−10^ and 6 × 10^−11^ m^2^/s. The broadening of the D-projection in D−T_2_ data at 2.5 MHz includes this bimodality but is not observed due to sensitivity loss.

In native crude oil, two distributions of slow and fast D are observed for CH_2_ and CH_3_ peaks, cantered at 0.30 and ~1.6 × 10^−10^ m^2^/s, indicating CH_2_ and CH_3_ are on the same aliphatic chain. The separation factors for these D are 5.3 for CH_2_ and 6.3 for CH_3_ in maltene.

Figure 12c demonstrates the variations of D with asphaltene concentration. The fastest D decreases linearly with asphaltene concentration, while the slowest one remains relatively unchanged, indicating two populations of hydrocarbons in crude oil. The slowest D is likely associated with long-chain hydrocarbons, and the fastest with short-chain hydrocarbons.

To overcome the not simple differentiation of the numerous components existing in crude oil, the potential of the multi-way statistical evaluation of the DOSY NMR spectra was recently tested by using the TUCKER3 model [70]. The advantage of the DOSY spectrum with regard to the simple proton spectrum is clearly demonstrated in both Figure 13 and Figure 14. As seen in Figure 13, DOSY spectra of crude oil samples display separated signals belonging to species with different diffusion properties. These signals are overlapped in the chemical shift dimension (Figure 13) and, thus, cannot be unambiguously identified. The same is true for the DOSY spectra of asphaltene samples isolated from the respective crude oils, as shown in Figure 15, where several aggregates can be recognized according to their different D coefficients. Although different components present in oil samples can be separated in the diffusion dimension by their mass and size, it is not straightforward to identify and differentiate which spectra belong to asphaltenes and which belong to other oil samples. Therefore, the DOSY spectra look almost the same, but the samples are significantly separated in the classification space.

Such multiway TUCKER3 decomposition was used for the first time to analyze the two-dimensional matrix of complex NMR data for petroleum samples, see Figure 16.

DOSY NMR spectra of 50 samples (45 for establishing the statistical model and 5 for its validation) were recorded and evaluated by an in-house-developed code incorporating multiway analysis to set up a tool for predicting their identity and origin. A statistical model was developed that can identify and separate asphaltene samples from those of crude oils, vacuum and atmospheric residues, and resins, which is not possible directly from the NMR spectra. Once the validity of the statistical decomposition model for the evaluation of DOSY spectra was tested on dozens of asphaltene samples, it was possible to identify and group crude oil samples of different origins.

^1^H DOSY NMR was used to estimate the size of asphaltene aggregates in toluene solutions, extracted from a light Arabian crude oil and to investigate the possible presence of occluded residues of maltenes in the asphaltene aggregates [37]. The authors did not apply DOSY NMR data by varying the asphaltene solution concentration, contrary to most studies on asphaltenes by DOSY NMR. However, DOSY results indicate the presence of nanosized aggregates and some remaining solvent molecules, some long alkyl chain species not chemically linked to the aromatic sheets, and the coexistence of archipelago and continental structural patterns.

To explore aggregate sizes and occluded compounds in the sample, a ^1^H DOSY spectrum was recorded (Figure 17). This spectrum for the asphaltene sample at 8 wt % reveals signals in the 1H aliphatic range (0.5–1.9 ppm) and shows bimodal peaks. The diffusion coefficients (D) for toluene range from 1.31 × 10^−5^ to 1.75 × 10^−5^ cm^2^/s, consistent with decreasing D as sample concentration increases. Aromatic and aliphatic protons from sample molecules have D values ranging from 5.80 × 10^−7^ to 2.66 × 10^−6^ cm^2^/s, indicating separation between toluene and sample molecules. Signals with D < 2.66 × 10^−6^ cm^2^/s are related to asphaltene aggregates, including nano-, micro-, and macroaggregates.

Most D values in Figure 17 (5.80 × 10^−7^ to 1.29 × 10^−6^ cm^2^/s) likely belong to nanoscale aggregates, consistent with the modified Yen model, which describes asphaltene nanoaggregates as having a rigid core with mobile alkyl chains. Aromatic signals for species with the lowest D values (5.80 × 10^−7^ cm^2^/s) are absent, aligning with the continental/island model of asphaltenes. As D increases to 6.71 × 10^−7^ cm^2^/s, aromatic protons appear, suggesting fewer fused aromatic rings or multiple aromatic cores. Higher D values (7.91 × 10^−7^ to 1.29 × 10^−6^ cm^2^/s) show no aromatic protons, possibly due to lower species concentration or longer alkyl chains, indicating coexisting continental/island and archipelago structures.

Vuković et al. discovered an interesting phenomenon related to the effect of the magnetic field of DOSY measurements on D of asphaltene aggregates, indicating higher molecular mobility and possible degradation of aggregates with increasing field strength [71].

Table 1 provides an overview of various types of 2D NMR measurements and their applications. It is important to note that while the applications listed in the table are representative of those commonly found in literature, they are not exhaustive.

## 4. Case Studies on Bitumen Characterization Using NMR

In the realm of bitumen research, the rheological properties of bitumen, particularly its behavior under various conditions, are of significant interest. One of the key techniques employed in this area is NMR, which has proven to be an invaluable tool in characterizing these properties. The following section presents a detailed analysis of specific studies where NMR was used to characterize different bitumen properties.

### 4.1. Bitumen Emulsion and Analysis of Physico-Mechanical Properties 

^1^H NMR

A notable study conducted by Filippov et al. investigated self-diffusion in a bitumen emulsion using ^1^H NMR [87]. The researchers found that the emulsion forms two phases: a continuous phase and a dispersed phase. The continuous aqueous phase primarily contains water, with the energy of activation of the diffusion process being equal to that of bulk water, while its diffusivity is smaller than that of bulk water by a factor of 2. On the other hand, the dispersed phase consists of bitumen droplets containing confined water, whose dynamics are characterized by a fully restricted diffusion regime in cavities with sizes of approximately 0.11 μm. Based on these findings, the researchers proposed that the studied bitumen emulsion can be described by a model of a complex multiple emulsion of the water/oil/water (WOW) type. This model aligns well with data from ^1^H NMR spectroscopy and diffusometry of the bitumen emulsion doped with paramagnetic MnSO_4_(aq), as well as with an additional ^1^H NMR study of the emulsion structure, in which emulsion stability was compromised by freezing at 253 K. This study demonstrates that NMR diffusometry can be effectively used to characterize “multiple emulsions”. It provides valuable insights into the structure and dynamics of bitumen emulsions, contributing to our understanding of their formation and stability. These findings have significant implications for the production and application of bitumen emulsions in various industries.

PFGSE-NMR

One key area where NMR spectroscopy has been extensively utilized is the study of bitumen’s molecular dynamics. Techniques such as NMR PFGSE and relaxometry, including CPMG spin-echo pulse sequence, allow researchers to probe the effects of additives and polymers on bitumen’s structure and behavior. These techniques provide valuable information on how different compounds interact within the bitumen matrix, influencing its mechanical and physico-chemical properties.

For instance, Caputo et al. (2019) investigated the mechanical and physico-chemical properties of a new kind of modified bitumen [88]. The bituminous binders were modified to understand the effect on the structural properties of several compounds such as a polymer elastomer as Styrene Butadiene Rubber (SBR), polymer thermoplastic polypropylene (PP), and a waste plastic (waste PP). The structure of both neat and modified bitumens was investigated by analyzing the mobility of the oily maltene fraction through PFGSE-NMR experiments. Although this research provides valuable insights into the mechanical performance of bitumen modified with waste PP and SBR additives, the PFGSE-NMR technique was successfully employed in bitumen composition characterization.

In a study by Jiang et al., the researchers focused on the characterization of water-in-diluted-bitumen emulsions using NMR measurements [89]. They used an NMR-restricted diffusion experiment (PFGSE) to measure the emulsion drop-size distribution. The data from PFGSE measurements suggested that the water-in-diluted-bitumen emulsion is very stable over time without an added coalescer. They also obtained the sedimentation rate of the emulsion and water droplet sedimentation velocity from NMR one-dimensional (1-D) T_1_ weighted profile measurement. The study found that PR 5, a polyoxyethylene (EO)/polyoxypropylene (PO) alkylphenol formaldehyde resin, is an optimal coalescer at room temperature. For the sample without fine clay solids, complete separation can be obtained; for the sample with solids, a rag layer forms between the clean oil and free water layers, preventing further coalescence and water separation.

Johns et al. applied NMR to quantify an emulsion droplet size distribution (DSD) via its ability to measure restricted molecular self-diffusion [90]. They described the methodology along with the advantages and limitations of the technique and elucidated recent highlights and typical applications. This study demonstrates the potential of NMR in providing detailed insights into the structure and dynamics of emulsions. These studies collectively highlight the versatility and efficacy of NMR techniques in providing detailed insights into the rheological properties of bitumen emulsions, particularly in evaluating their stability, drop-size distribution, and oil and water content. These insights are crucial for improving the performance and longevity of bitumen emulsions in various applications.

PFGSE-NMR combined with 1D-imaging

NMR is a powerful technique to characterize the molecular composition of shale rock, which contains both producible hydrocarbon fluid and semi-solid organic solids such as bitumen. Bitumen is a complex mixture of organic molecules that can affect the NMR signal and the interpretation of the relaxation times. Therefore, different methods have been proposed to separate the bitumen signal from other components, such as water, clay, and kerogen, in shale samples.

A method is to apply a simultaneous Gaussian-Exponential (SGE) inversion method to the CPMG data, which can produce more physically realistic results than the ILT and display more consistent relaxation behavior at high magnetic field strengths [91]. The SGE inversion can also reduce the signal overcall at short T_2_ times and the residuals compared to the inverse Laplace method. The SGE inversion can be applied to various oil shale samples and other fields where the sample relaxation consists of both Gaussian and exponential decay, such as material, medical, and food sciences.

A study conducted by Remi et al. exemplifies the application of NMR in this context [42]. The authors investigated the infiltration of water into porous matrices composed of bitumen and salts with different solubilities. They used a combination of PFGSE-NMR and Environmental Scanning Electron Microscopy (ESEM) to monitor the evolution of the porous structure over time, due to water infiltration. This approach allowed them to observe the slow seeping of water into the material and understand how the leaching behavior varies depending on the type of salts dispersed inside the bitumen.

Furthermore, they performed NMR relaxation and diffusion measurements, which when combined with 1D imaging, provided information on the surface-to-volume ratio of the water-filled porous network at different times as a function of depth. This research provides valuable insights into the leaching behavior of porous bitumen–salt matrices and could serve as a basis for modeling their evolution on longer timescales.

In a parallel study conducted by Le Feunteun et al., the researchers demonstrated that coupling NMR 1D-imaging with the measurement of NMR relaxation times and self-diffusion coefficients can be a powerful approach to investigating fluid infiltration into porous media [92]. They studied the slow seeping of pure water into hydrophobic materials, specifically three model samples of nuclear waste conditioning matrices. These matrices consisted of a dispersion of highly soluble NaNO_3_ and/or poorly soluble BaSO_4_ salt grains embedded in a bitumen matrix.

The researchers went beyond studying the moisture progression according to the sample depth. They analyzed the water NMR relaxation times and self-diffusion coefficients along its 1D concentration profile to obtain spatially resolved information on the solution properties and on the porous structure at different scales. Interestingly, they found that when the relaxation or self-diffusion properties are multimodal, the 1D profile of each water population is recovered.

Three main levels of information were disclosed along the depth profiles: (i) the water uptake kinetics, (ii) the salinity and the molecular dynamics of the infiltrated solutions, and (iii) the microstructure of the water-filled porosities, which consisted of open networks coexisting with closed pores1. All these findings were fully validated and enriched by NMR cryoporometry experiments and environmental scanning electronic microscopy observations. Surprisingly, the results clearly showed that insoluble salts enhance the water progression and thereby increase the capability of the material to uptake water.

Comparing the two studies, both Remi et al. (2019) and Le Feunteun et al. (2011) utilized NMR techniques to investigate the infiltration of water into porous bitumen matrices. However, while Remi et al. focused on the leaching behavior of bitumen–salt matrices, Le Feunteun et al. provided a more detailed analysis of the fluid infiltration process, including the water uptake kinetics, the salinity and molecular dynamics of the infiltrated solutions, and the microstructure of the water-filled porosities [92].

These studies collectively highlight the versatility and efficacy of NMR techniques in providing detailed insights into the rheological properties of bitumen.

### 4.2. Bitument Content and Molecular Composition Determination

In the comprehensive review by Elsayed et al. [93], the authors delve into the wide-ranging applications of NMR in the oil and gas industry. The review encompasses both laboratory and field-scale measurements, providing a holistic view of the current state of NMR applications in the industry. This includes the characterization of bitumen, among other substances, highlighting the versatility and efficacy of NMR techniques in providing detailed insights into the molecular composition and physical properties of such complex systems. The review serves as a valuable resource for researchers and industry professionals alike, offering a thorough understanding of the potential and capabilities of NMR in the oil and gas industry.

NMR spectroscopy has emerged also as a powerful tool for studying the content, molecular composition, and dynamics of bitumen systems, offering unique insights into their complex nature. In addition to traditional techniques, such as infrared spectroscopy and chromatography, NMR provides a non-destructive and versatile approach to characterizing bitumen at the molecular level. This capability has opened up new avenues for understanding bitumen’s properties and behavior in various applications, including road pavement construction and oil extraction.

Remarkably, the content and group composition of bitumen in rock samples can be determined by NMR techniques without the need for extraction. One such technique is based on a combination of FID and modified CPMG measurements, which was developed by Ranel et al. [31]. This technique allows obtaining the SARA fractions of bitumen directly in the porous media and has been shown to produce results that are close to conventional SARA analysis (ASTM D4124-09) [94]. However, conventional SARA analysis may be affected by the incomplete extraction of bitumen from the mineral matrix, as revealed by micro-X-ray tomography (micro-XCT) and focused ion beam combined with scanning electron microscope (FIB-SEM) imaging. Therefore, the NMR technique may offer a more accurate and convenient way to characterize the bitumen entrapped in rocks.

LF-NMR Relaxometry

In addition to the methods previously discussed, another patented technique further expands the scope of NMR in bitumen characterization. This method focuses on determining the composition of a sample that includes heavy oil or bitumen and water, utilizing LF-NMR. The process involves capturing the NMR spectrum of the sample at both relatively low and high temperatures. The content of oil or water is then deduced from these spectra and the differential spectrum [95]. This innovative approach enhances the accuracy and efficiency of characterizing bitumen and heavy oil samples, thereby contributing valuable insights into their composition and properties.

In another study [96] NMR relaxometry was successfully tested as a tool for accurately measuring the oil and water content of streams with and without emulsions present in the samples. The method proved to be at least as good as conventional extraction methods (i.e., Dean-Stark). The technology was tested with both artificially and naturally occurring emulsified streams with an accuracy better than 96%. This promising result led to the design of an online NMR relaxometer for oil/water stream measurements under the conditions encountered in the production of heavy oil and bitumen. The LF-NMR method, applied at two frequencies of 2 and 5 MHz, was exploited by Zielinski et al. to study the aggregation of asphaltenes in both model solutions and crude oil [97,98].

FID and CPMG-NMR

One method is to apply combined FID and CPMG measurements and interpret the spliced data using a combined Gaussian and exponential inversion method [40]. This method assumes that the bitumen signal has a Gaussian distribution in the transverse relaxation time (T_2_*) domain, while the water signal has an exponential distribution in the spin-spin relaxation time (T_2_) domain. The Gaussian component of the resultant T_2_*|T_2_ distribution is independent of moisture content and scales with the total organic content of the shale cores, while the exponential component scales linearly with the moisture content. This method can be applied at low magnetic field strengths (20, 40, and 60 MHz), which are suitable for bench-top NMR spectrometers.

LF-NMR relaxometry was developed by Volkov et al. by using all experimental points of FID and CPMG signal to numerically determine the bitumen content and analyze its heterogeneous components [99]. This technique made it possible to quantitatively determine the content of not only asphaltenes but also resins, aromatics, and saturates in heavy oils. In addition, the method was tested on a wide range of molecular weights from light to extra heavy oils, and a high correlation of resins and asphaltene content was obtained, respectively, by NMR and SARA analysis (ASTM D4124-09) [100]. The composition of bitumen entrapped in rock samples was successfully determined by applying a combination of FID and modified CPMG measurements using the LF-NMR relaxometry technique [31]. Good agreement of the NMR results with thermogravimetric analysis (TGA) and pyrolysis was observed, demonstrating the possibility of estimating the composition of the SARA fraction directly in the rock without resorting to destructive methods.

TD-NMR

Furthermore, a patented method provides a novel approach for quantifying bitumen and/or water in a sample comprising bitumen, water, and solids using a TD-NMR pulse spectrometer. The method involves initially saturating the magnetization of the sample so that essentially no magnetization remains in the +Z axis. The sample is then subjected to a sequence of RF pulses optimized for the measurement of bitumen and water in the sample. The transverse relaxation (T_2_) echo trains are recorded after incremental longitudinal relaxation to produce a raw TD-NMR data set for the sample. The amount of bitumen and water is determined by means of a partial least square optimization-based chemometric model. This model relates TD-NMR data sets obtained from a training set of samples comprising bitumen, water, and solids to the training samples’ corresponding reference values obtained from a standard analysis method for determining bitumen and water [101]. This patented method offers a robust and efficient approach to bitumen and water quantification in complex samples.

TLC and ^1^H-NMR

Building on the use of NMR in bitumen research, a study by Oliviero Rossi et al. developed a technique to investigate the composition of bitumens [32]. This technique uses TLC to separate the different fractions, and NMR spectroscopy to assess and quantify the aliphatic hydrogen with respect to the aromatic part. ^1^H-NMR analysis was conducted in solution, using CCl_4_ as a solvent, on three different neat bitumens and on their asphaltene and maltene fractions. The combined application of TLC and ^1^H-NMR spectroscopy enables the advanced characterization of bitumens supplied from different sources or obtained from different processes. This further allows addressing the use of specific modifications according to the bitumen’s final applications. This study underscores the versatility of NMR in providing detailed insights into the molecular composition of bitumen.

Further expanding on the use of NMR in bitumen research, Teltayev et al. discussed the quantitative determination of the fragmentary composition of road bitumen of grade BND 100/130 and its components (asphaltenes, resins, and oils) using NMR spectroscopy [43]. The group chemical composition of the bitumen was determined by the adsorption chromatography method. It was identified that the bitumen and its components consist only of aromatic and aliphatic protons, which account for 2.4–10.2% and 9.8–97.6%, respectively. The study found no presence of olefinic elements in them. The majority (79–81%) of nuclei of carbon atoms relate to quaternary carbon atoms of saturated compounds. Primary carbon atoms in the methylene group (CH_2_) are contained in the least quantity: bitumen—1.32%; asphaltenes—0.6%; resins—3.24%; and oils—2.11%. Primary carbon atoms, linked with the CH-group or aromatic nucleus, occupy an intermediate position and are contained in the quantity of 17–20%. This research provides a detailed insight into the molecular composition of bitumen and its components, further enhancing our understanding of bitumen characterization using NMR techniques.

2D NMR

In a study by Warkovits et al. (2022), the researchers utilized NMR techniques to analyze the structural and chemical peculiarities of bitumen and its polarity-based fractions [39]. They employed a 2D NMR technique, heteronuclear multiple bond coherence (HMBC), which revealed minor contributions of carboxylic esters or amides within the resins. Hydroxyl and amino groups were analyzed by ³1P NMR after derivatization, uncovering minor amounts of alcohols, amides, and carboxylic acids in the most polar fractions. Furthermore, they used DOSY to estimate average molecular sizes and weights. By applying in-depth NMR analysis to bitumen and combining it with complementary analytical techniques such as infrared/fluorescence spectroscopy and elemental analysis, they provided a unique foundation for future aging studies. This work demonstrates how multi-spectroscopy can offer new insights into the structure and chemistry of bitumen.

Combined NMR-FTIR

Rakhmatullin et al. showed that high-resolution NMR and FT-IR spectroscopy experiments were applied to obtain detailed information on the hydrocarbon chemistry of three light and three heavy crude oils [38]. The researchers determined quantitative fractions of aromatic molecules and functional groups constituting oil hydrocarbons using ^13^C NMR spectroscopy. They performed a comparative analysis of the oil samples with different viscosity, origin, and preliminary treatment. The study provided insights into the SARA composition and important information about aromaticity, oxidation behavior, branching, aliphaticity, and sulfurization of the studied oil samples. The integral characteristics of high-resolution NMR and FT-IR spectra demonstrated great potential to study the structure and characterization of light and heavy crude oils, potentially substituting traditional fractionation procedures. The relationships between spectroscopic parameters obtained by high-resolution NMR and FT-IR spectroscopy methods and crude oil compositions can be useful for fast prediction of crude oil properties due to different types of treatment, including thermal methods for enhanced oil recovery. The study also suggested that quantitative proportions of functional groups obtained by NMR and spectral indices obtained by FT-IR can be one of the criteria for developing a fingerprint approach. This research underscores the utility of ^1^H and ^13^C NMR spectroscopy as a well-recognized technique for establishing structural formulas and the spatial and electronic structure of individual organic compounds either first synthesized or isolated from natural raw materials.

In conclusion, NMR spectroscopy offers a powerful and versatile tool for the study of bitumen, providing detailed insights into its molecular composition and physical properties. The use of techniques such as FID, CPMG, PFGSE, and LF-NMR, combined with the incorporation of various additives, allows for a comprehensive understanding of bitumen and its potential applications.

### 4.3. Bitumen Ageing

In a study by Khavandi et al., the researchers explored the impact of poly 2-hydroxyethyl methacrylate (PHEMA) on the aging process of pure bitumen [102]. They incorporated different percentages of PHEMA (3%, 5%, and 7% by weight) into the bitumen and subjected it to short- and long-term aging using the Rolling Thin Film Oven (RTFO) and Pressure Aging Vessel (PAV) tests. The physical and chemical structure of the aged bitumen samples was then analyzed using softening point tests and FT-IR and NMR techniques. The results demonstrated that PHEMA effectively delays the aging process and reduces the aging indexes of bitumen.

FT-NMR-SDC

In a separate study, Oliviero Rossi et al. employed Fourier Transform Nuclear Magnetic Resonance Self-Diffusion Coefficient (FT-NMR-SDC) spectroscopy as a novel method to investigate and compare the microstructural changes between virgin (pristine) and aged bitumen [103]. The study revealed that FT-NMR-SDC spectroscopy can effectively discern the differences in the molecular mobility and the size distribution of the bitumen fractions. This research underscores the potential of NMR techniques in providing detailed insights into the aging process of bitumen.

These studies collectively highlight the versatility and efficacy of NMR techniques in evaluating its aging process during various stages such as heating, production, storage, transport, laying, and compaction of hot mix asphalt and asphalt pavement. They also shed light on the potential of using additives such as PHEMA to modulate the aging process of bitumen. These insights are crucial for improving the performance and longevity of bitumen in various applications.

In a study by Caputo et al., the researchers demonstrated that the modification of maltenes to asphaltenes is responsible for the aging character of bitumen [104]. They used FT-NMR-SDC spectroscopy to confirm that the asphaltene content increases during the aging process. This technique showed a strong ability to monitor the aging processes and to highlight the structural differences induced during aging. The Atomic Force Microscopy (AFM) results indirectly confirmed what was obtained via the FT-NMR-SDC technique by showing that the size of asphaltenes in aged bitumens is increased when compared with the size of asphaltenes in unaged bitumen.

ILT NMR

In an additional study by Caputo et al., the researchers showed that Inverse Laplace Transform (ILT) is particularly useful when the signal is characterized by multi-exponential decay [105]. This is often the case in spin relaxation or in the dephasing of the NMR spin echo signal associated with supra molecular aggregation under the influence of pulsed magnetic or internal field gradients. As a novel approach to observe the real rejuvenating effect of the potential additive, an ILT of the experimental NMR spin-echo decay (T_2_) was applied. The potentialities of a new, non-toxic, and eco-friendly biocompatible additive on aged bitumen were explored for the first time as a bitumen rejuvenator by means of advanced rheological and Relaxometry-NMR measurements. The new rejuvenator helps to rearrange the structure of the aged bitumen (aiming at the original one), and this mechanism can be observed by ILT NMR analysis.

Comparing these studies, both sets of researchers utilized NMR techniques to investigate the aging process of bitumen. However, while the first study focused on the modification of maltenes to asphaltenes and the increase in asphaltene content during the aging process, the second study introduced a novel approach using ILT to observe the rejuvenating effect of a potential additive on aged bitumen. These studies collectively highlight the versatility of NMR techniques in providing detailed insights into evaluating its aging process and the potential of using additives to modulate this process.

## 5. Comparison between NMR Techniques Used in Bitumen Analysis

As has been widely documented in the literature, the NMR techniques used to characterize complex colloidal systems such as bitumen have provided many unique and valuable insights on both its structural and dynamic properties. However, the selection of NMR methods for characterizing bituminous materials is influenced by several key factors. Firstly, NMR spectroscopy is crucial for establishing structures and compositions of organic compounds in petroleum products, with a focus on hydrocarbons, due to its ability to correlate integral intensities of signal groups with molecular fragments in ^1^H and ^13^C NMR spectra. In particular, NMR spectra are usually acquired to determine the content and group composition of bitumen to assess and quantify the aliphatic hydrogen part with respect to the aromatic part. In fact, due to the complex nature and composition of bitumen, it is difficult to distinguish the discernible differences found in bituminous systems with similar physical properties [32,39,43]. Indeed, NMR techniques, such as 1D- and 2D-NMR, are valuable for studying the molecular architecture and aggregation processes of polar components such as asphaltenes in bituminous materials, providing essential data for comprehensive structural characterization (Oliviero Rossi et al., 2018 [32]; Vuković et al., 2019 [34]).

Secondly, dynamics information such as, e.g., molecular mobilities within bituminous matrices, can be investigated through the NMR diffusometry technique for the determination of self-diffusion coefficients of supramolecular aggregates inside cavities.

For example, PFGSE-NMR is used in bitumen research to monitor the infiltration of water into porous bitumen–salt matrices (due to a leaching phenomenon) and characterize the evolution of the porous structure. In fact, this technique enables performing 1D-imaging of water at different times to monitor its slow seeping inside the material. It is shown that coupling 1D-NMR imaging with the measurements of relaxation times and self-diffusion coefficients can be a very powerful approach to investigating fluid infiltration into porous media. The relaxation and diffusion measurements performed in combination with 1D-imaging yield information on the surface-to-volume ratio of the water-filled porous network, at different times, as a function of depth. In the case of the matrix-containing insoluble salts, relaxation measurements lead to discrimination between two different water populations differing by their T_2_ or T_1_ values [42,97,98,106].

NMR relaxometry is also used in bitumen research to evaluate the aging process during heating, production, storage, and transport, as well as when laying Reclaimed Asphalt Pavement (RAP) and during the compaction of hot mix asphalt and asphalt pavement. Bitumen aging encompasses volatilization and oxidation, which enable changes in the material molecular structure. A real rejuvenator helps to rearrange the colloidal structure of the oxidized bitumen, thus recreating one similar to the fresh bitumen. Through NMR relaxation measurements, it is possible to evaluate the rejuvenation effects of aged bitumen upon the addition of specific additives by applying the ILT to the NMR spin-echo decay (T_2_). ILT is particularly useful when the signal is characterized by multi-exponential decay, for example in spin relaxation or in the dephasing of the NMR spin echo signal associated with supramolecular aggregation under the influence of pulsed magnetic fields or internal field gradients. The rejuvenator additive helps to rearrange the structure of the aged bitumen (aiming at the original one), and this mechanism can be observed by ILT-NMR analysis. The self-diffusion coefficients give information about the mobility of organic compounds in petroleum products and the microstructure of bitumen [99,101,102]. The molecular dynamics of asphalt binders can be investigated by NMR PFGSE and relaxometry (CPMG spin-echo pulse sequence) to capture the effect of the polymers on the bitumen system [87]. One method is based on a combination of FID and modified CPMG measurements using an LF-NMR relaxation technique. It allows obtaining the content and group composition (SARA fractions) of bitumen directly in the porous media without extraction [31,107]. A method is provided for quantifying bitumen and/or water in a sample comprising bitumen, water, and solids using a pulsed TD-NMR spectrometer. It includes the initial step of flipping the magnetization of the sample so that essentially no magnetization remains in the +Z-axis. Next, the sample is irradiated with an RF pulse sequence optimized for measuring bitumen and water in the sample. Finally, trains of transverse relaxation (T_2_) echoes are recorded after incremental longitudinal relaxation to produce a raw TD-NMR data set for the sample. Thanks to this method, the amount of bitumen and water can be determined using a chemometric model based on partial least squares optimization that relates the TD-NMR data sets obtained from a set of training samples including bitumen, water, and solids to the corresponding reference values of training samples obtained from a standard analytical method for the determination of bitumen and water [101,108]. Among the NMR techniques applied to the study of bitumen, relaxometry is commonly used to provide lithology-independent porosity and pore size estimates for petroleum resource evaluation based on fluid-phase signals. However, in shales, substantial hydrogen content is associated with solid and fluid signals, and both may be detected. Depending on the motional regime, the signal from the solids may be best described using either exponential or Gaussian decay functions. When the ILT—the standard method for analysis of NMR relaxometry results—is applied to data containing Gaussian decays, this can lead to physically unrealistic responses such as signal or porosity overcall and relaxation times that are too short to be determined using the applied instrument settings. A SGE inversion method can be used to simulate data and measured results obtained on a variety of oil shale samples. The SGE inversion produces more physically realistic results than the ILT and displays more consistent relaxation behavior at high magnetic field strengths. Residuals for the SGE inversion are consistently lower than for the inverse Laplace method and signal overcall at short T_2_ times is mitigated. Beyond geological samples, the method can also be applied in other fields where the sample relaxation consists of both Gaussian and exponential decay, for example, in material, medical, and food sciences [40,95,96].

^1^H NMR proves to be a powerful technique for studies of bituminous emulsions. In fact, the emulsion forms two phases: continuous and dispersed. The continuous aqueous phase contains mainly water, with the energy of activation of the diffusion process equal to that of bulk water, while its diffusivity is smaller than that of bulk water by a factor of 2. The dispersed phase consists of bitumen droplets containing confined water, whose dynamics are characterized by a fully restricted diffusion regime in cavities with sizes of ~0.11 μm. Therefore, the bituminous emulsion can be described by a model of a complex multiple emulsion of the water/oil/water (WOW) type. Indeed, the characterization of water-in-diluted-bitumen emulsions and the transient behavior of emulsions undergoing phase separation can be studied by PGSE-^1^H NMR experiments able to determine the emulsion droplet size distribution (DSD) from the measurement of restricted molecular self-diffusion [109]. Moreover, the sedimentation rate of emulsion and water droplet sedimentation velocity can be obtained from NMR one-dimensional (1-D) T_1_ weighted profile measurements, while the emulsion flocculation can be deduced by comparing the sedimentation velocity from experimental data with a modified Stokes’ Law prediction [104]. During production operations in heavy oil and bitumen formations where thermal recovery methods are applied, the fluids produced are often in the form of emulsions. This is also true in non-thermal recovery methods whenever oil and water are coproduced but at a lower degree of severity. Conventional flow measuring devices are capable of measuring oil and water streams when they are segregated, but they fail when oil-in-water or water-in-oil emulsions form. Conventional methods are also not reliable when there are solids flowing in the stream. Thus, Low-field (LF) NMR relaxometry can be used as a tool for accurately measuring the oil and water content of such streams with and without emulsions present in the samples [89]. Overall, the multifaceted nature of bituminous materials and the need for detailed structural information drive the selection and application of NMR methods in their characterization.

## 6. Challenges and Limitations

The characterization of bitumen by using NMR spectroscopy presents challenges due to the extreme complexity of bitumen’s molecular structure, particularly with components such as asphaltenes. Indeed, the wide variety of chemical and rheological properties of bitumen is complex, especially considering the changes that occur during aging and the effect that different additives have on the structure of the bitumen itself [110]. While NMR spectroscopy is a valuable tool for studying organic compounds in petroleum products, the interpretation of NMR spectra of complex mixtures such as bitumen can be difficult [111]. Therefore, when using NMR spectroscopy as an analytical technique, the use of high-purity samples is mandatory. Additionally, the molecular complexity of bitumen makes it challenging to fully identify all compounds present, further complicating the analysis using NMR techniques [112]. Despite these challenges, NMR spectroscopy still remains a useful and very powerful method for providing valuable data on bitumen’s molecular architecture and aggregation processes, especially when combined with statistical methods such as a Bayesian estimation method for analyzing NMR data [113], support vector regression (SVR) [114], multiple linear regression (MLR) [115], and partial least square PLS [116], also associated with PLS mid-level data fusion [117].

## 7. Future Directions

Potential improvements in NMR techniques for bitumen characterization involve advancements in studying structural micro-modifications during bitumen aging [118]. Utilizing FT-NMR-SDC spectroscopy has shown promise in comparing virgin and aged bitumen, highlighting changes in maltenes to asphaltenes during the aging process [112]. Additionally, single-sided NMR relaxometry with the NMR-MOUSE has been successfully employed to assess rubber fraction and pavement heterogeneity in rubber-modified asphalt non-destructively, showcasing a significant step towards enhanced quality control of asphalt pavements [119]. These techniques provide valuable insights into the molecular architecture, aggregation processes, and content changes in bitumen, offering a more comprehensive understanding of bitumen properties and aging mechanisms for future research and development in the field of petroleum chemistry.

In the field of bitumen research, emerging trends and future research directions for NMR techniques include the use of NMR tools for assessing the composition of bitumens through techniques such as TLC and 1H-NMR spectroscopy to quantify different fractions and understand the chemo-mechanical properties of bitumens [95]. Additionally, NMR is being applied to identify reservoir bitumen in oil reservoirs by analyzing short transverse relaxation times (T_2_) and NMR porosity to calculate the saturation of reservoir bitumen, enabling more reliable reservoir evaluation and production planning [35]. Furthermore, the potential of NMR in quantifying asphaltene precipitation in crude oil is highlighted as a cost-effective and time-saving alternative to traditional methods, showcasing its potential for routine analysis in the industry. These trends indicate the growing importance of NMR in enhancing the understanding and application of bitumen in various industrial processes.

Finally, NMR can be utilized in feature engineering to enhance machine learning models to predict crude oil stability based on DOSY NMR spectra and other measured properties [120] for estimating unconventional reservoirs, enabling the selection of relevant features in spectra to simplify interpretation, reduce dimensionality, and improve data compatibility, as demonstrated in the analysis of bitumen from different regions [121,122].

We cannot conclude this paragraph without giving some hints about the possibility of exploiting the knowledge of bituminous materials and their constituents to go beyond, exploring other systems that share some characteristics with bitumens. For example, asphaltene adsorption and wettability dynamics in siliceous systems have been explored [85]. In this case, needless to say, NMR has been chosen as a technique of election to probe this problem. We believe that the knowledge in bituminous materials and in NMR can be of inspiration for future synergic work and research.

## 8. Conclusions

This work shows how NMR spectroscopy can play an important role in the road pavement industry by providing valuable insights into the composition and structural changes of materials such as bitumen and asphalt. Despite the inherent difficulties in analyzing such complex systems due to the extreme variety of molecules comprising bitumen and consequently the self-assembly governing their intermolecular structure, NMR can be a powerful tool, offering a wide variety of strategies of analytical approaches.

NMR techniques can, therefore, among all the other things, quantitatively determine the composition of road bitumen, assess the heterogeneity of rubber-modified asphalt pavements, characterize the molecular architecture of asphaltene components, evaluate the quality of pavements, monitor the evolution of structural changes in materials over time, and gain a deeper understanding of the complex mixtures present in asphaltene samples. In addition, recent trends are enhancing the power of this technique by synergically coupling the NMR investigation with statistical analysis, modeling, and other spectroscopies.

All this makes this method quite advanced and significant in the road pavement industry: it aids in quality control and paves the way for potential modifications and enhancements in road construction materials. All the described NMR techniques collectively contribute to a comprehensive understanding of asphalt binder properties and behavior, aiding in the development of high-quality and durable pavement materials.

It is true that NMR techniques can be somewhat cumbersome and hard to understand for non-experts in this type of spectroscopy. For this reason, this work has been thought to make NMR spectroscopy more accessible and understandable to researchers and professionals involved in various fields of investigation of bitumens and related materials. We achieved this by (i) showing the different pulse sequences characterizing and differentiating the most widespread NMR methods, which is the key factor to understand the philosophy of exploiting NMR techniques, and (ii) by presenting and comparing several cases of studies, rationally grouped according to the aspects to be unveiled in bituminous materials (bitumen physico-mechanical properties, bitumen content, and molecular composition, aging). In this way, we sincerely hope that this work can be of inspiration for researchers involved in bituminous materials, as well as in other complex systems, in planning new experiments and in producing novel thoughts for research of ever-increasing added value.


## Figures and Tables

**Figure 1 molecules-29-04038-f001:**
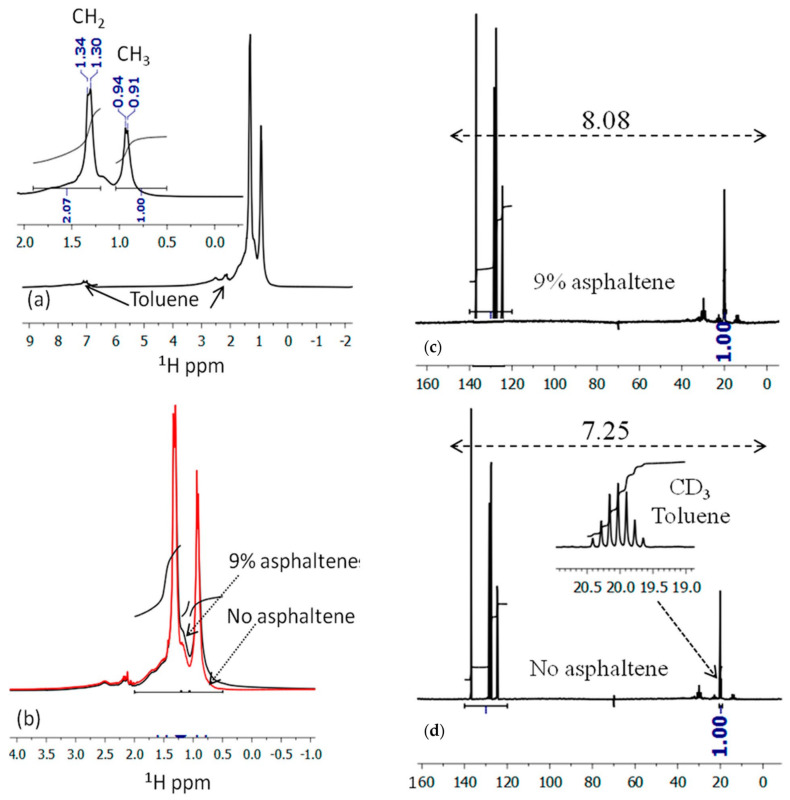
(**a**) 1H NMR spectrum at 300 MHz of crude oil with 9% asphaltene, with deuterated toluene d8 added. CH3 and CH2 peaks are assigned at 0.9 and 1.3 ppm, respectively, with a 2.07 integration ratio shown for CH2 relative to CH3 peaks. (**b**) Comparison of crude oil NMR spectrum with and without asphaltene. 13C NMR spectrum of crude oil at 600 MHz with 9% asphaltene (**c**) and 0% asphaltene (**d**). The inset shows the septuplet around 20 ppm due to 13C coupling with three methyl deuterons. Relative intensities of total carbons to the septuplet are indicated [7].

**Figure 2 molecules-29-04038-f002:**
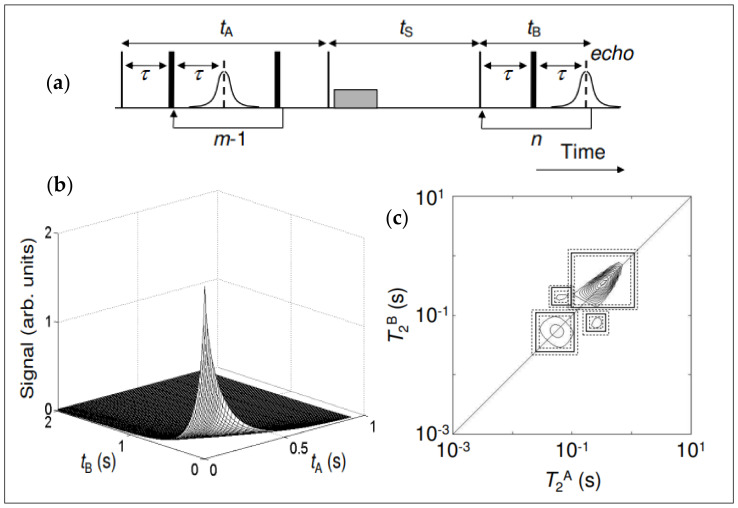
(**a**) REXSY (T_2_-T_2_) exchange measurement pulse sequence with 90° and 180° RF pulses, CPMG echo trains in *t*_A_ and *t*_B_ intervals, and storage time *t*_S_ for exchange observation. (**b**) Raw data from a T_2_-T_2_ exchange experiment with *t*_S_ = 750 ms and an S/N ratio better than 10^4^. (**c**) T_2_-T_2_ spectrum (*t*_S_ = 2 s) with manually defined peak integral boundaries (solid rectangles). Error in integration is estimated by adjusting boundaries by one relaxation time division (dotted rectangles) [58].

**Figure 3 molecules-29-04038-f003:**
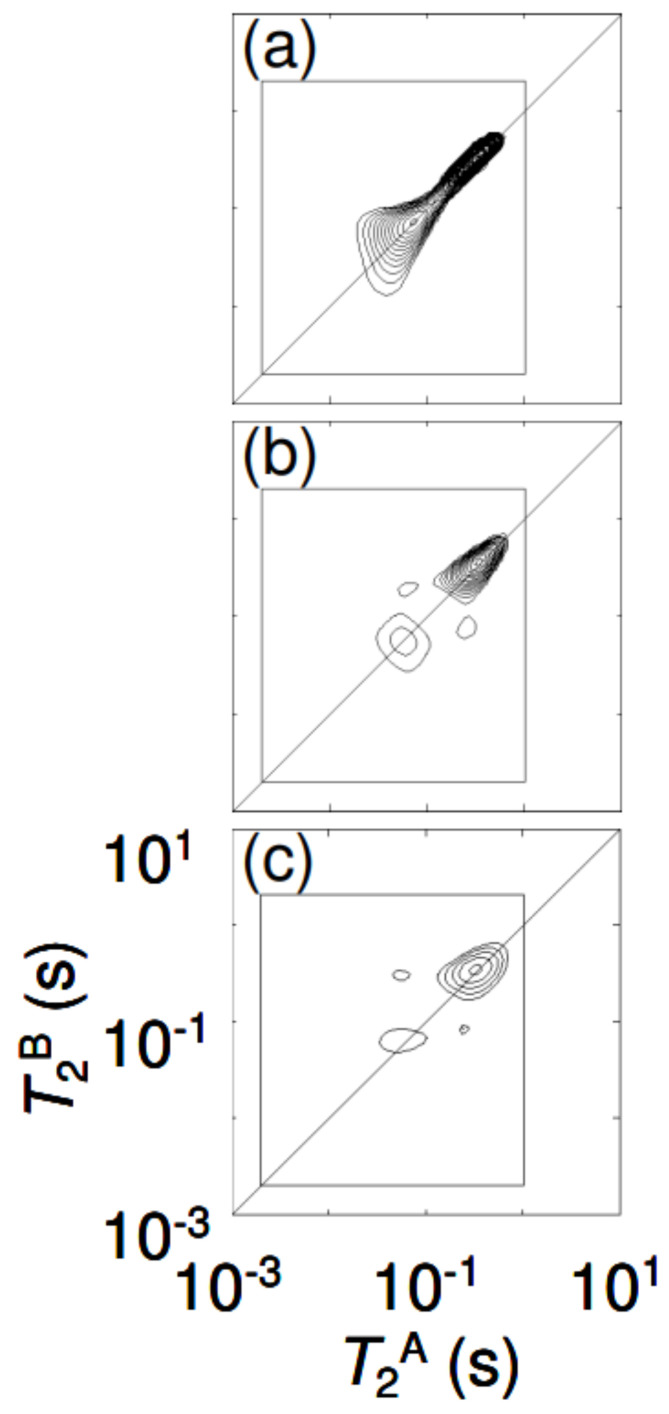
T_2_-T_2_ exchange spectra from water-saturated glass spheres in the sample, with storage delays of (**a**) 50 ms, (**b**) 2 s, and (**c**) 4 s [54].

**Figure 4 molecules-29-04038-f004:**
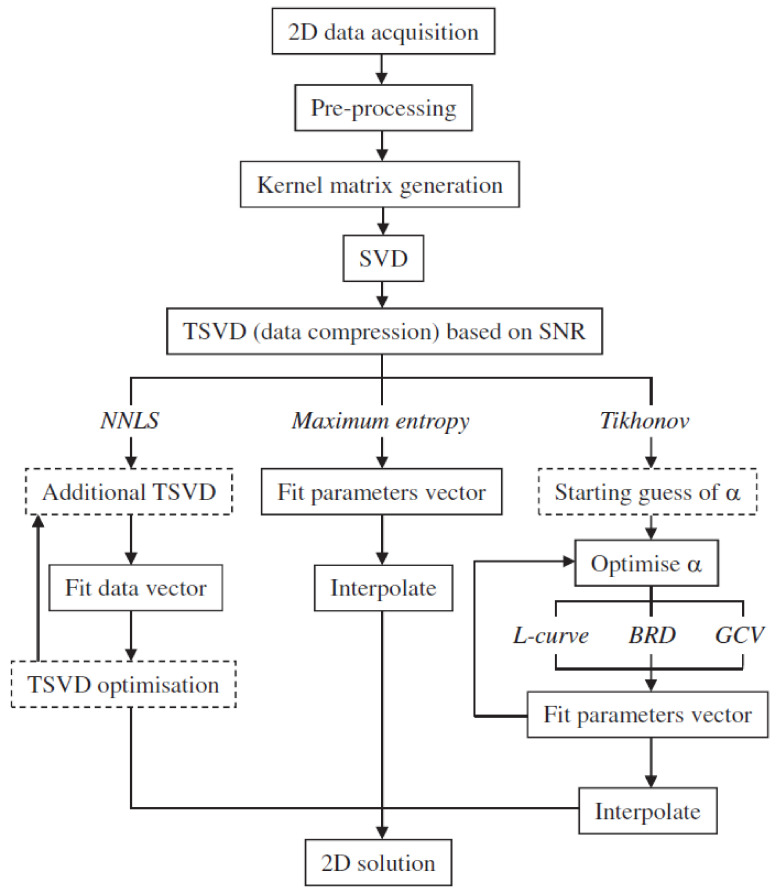
Flow chart indicating the key stages in the inversion of 2D relaxation or diffusion data. Operation names are shown in italics. Critical stages are surrounded by solid lines; optional stages are surrounded by dashed lines. An ’interpolate’ stage refers to the calculation of the 2D solution from a fitted set of parameters [60].

**Figure 5 molecules-29-04038-f005:**
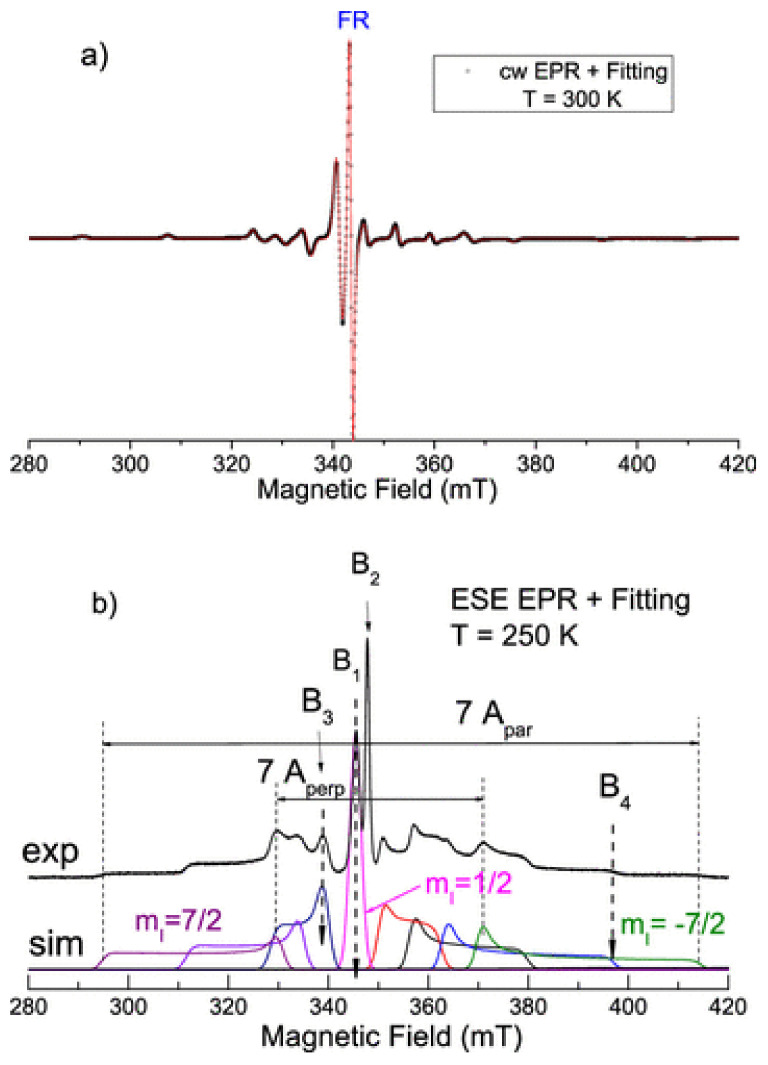
(**a**) CW EPR spectrum taken at T = 300 K and its fit by the parameters given in the text. The position of the single line of FR is marked. (**b**) Field-swept ESE-detected EPR spectrum at T = 250 K (exp) and simulation (sim) of the VO^2+^ spectrum. Partial contributions from the electron–nuclear transitions with the defined mI are shown and marked in color. Arrows indicate the values of the magnetic field (B_1_–B_4_) in the vicinity of which the EPR saturation curves and detailed DNP spectra were measured. B_1_, B_3_, and B_4_ correspond to VO^2+^ EPR transitions while B_2_ corresponds to FR EPR transition [61].

**Figure 6 molecules-29-04038-f006:**
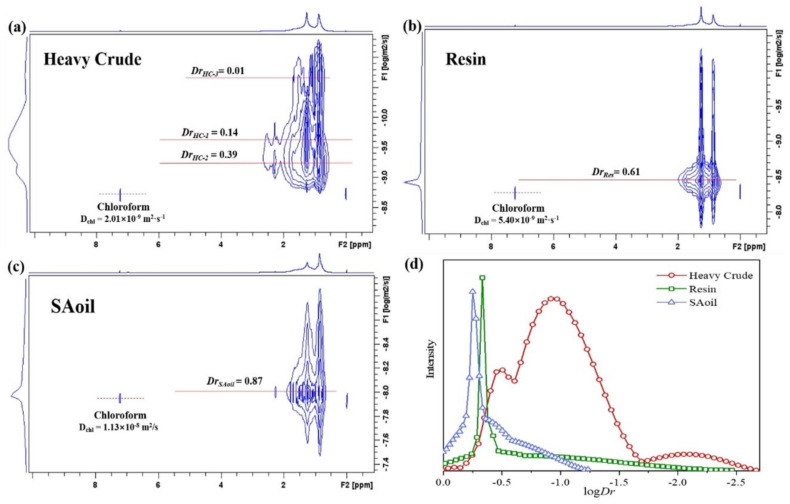
^1^H DOSY spectra of (**a**) Heavy Crude, (**b**) Resin, and (**c**) SAoil in deuterated chloroform solvent at the same concentration as 120 mgmL^−1^. The ^1^H chemical shift spectra and the projections of the diffusion maps on the diffusion dimension are attached on the top and the left to each ^1^H DOSY spectrum, respectively. (**d**) Comparison of the diffusion projection curves as a function of the logarithmic relative diffusivities (log D_r_) [64].

**Figure 7 molecules-29-04038-f007:**
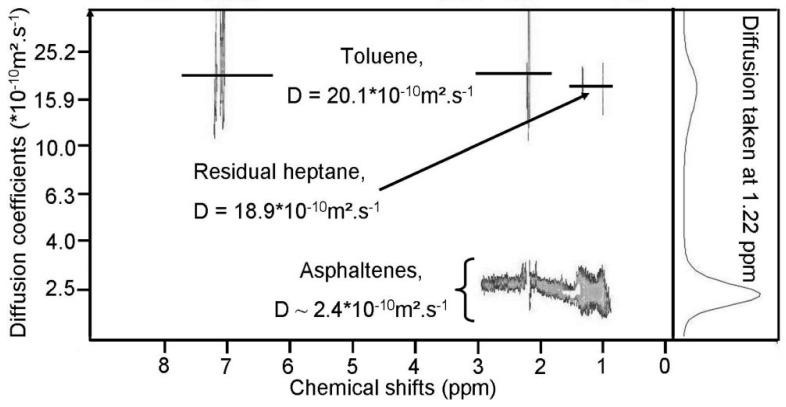
Asphaltene ^1^H DOSY spectrum in toluene and, in the right panel, the projection of a diffusion spectrum taken at 1.22 ppm [65].

**Figure 8 molecules-29-04038-f008:**
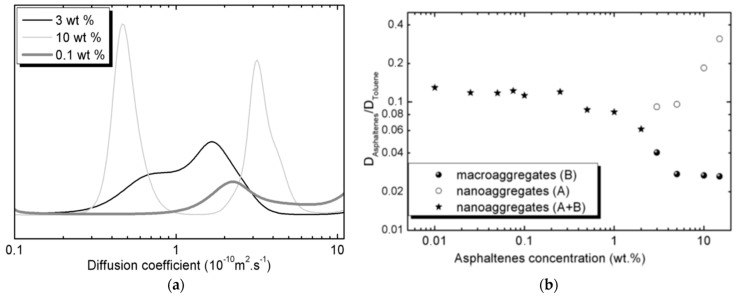
(**a**) Distribution of diffusion profiles extracted from the ^1^H DOSY spectra of Buzurgan asphaltenes analyzed at 0.1 (amplified 10 times), 3, and 10 wt % in toluene-*d*_8_. (**b**) Relative diffusivity of Buzurgan asphaltenes in toluene-d_8_ as a function of solute concentration [66].

**Figure 9 molecules-29-04038-f009:**
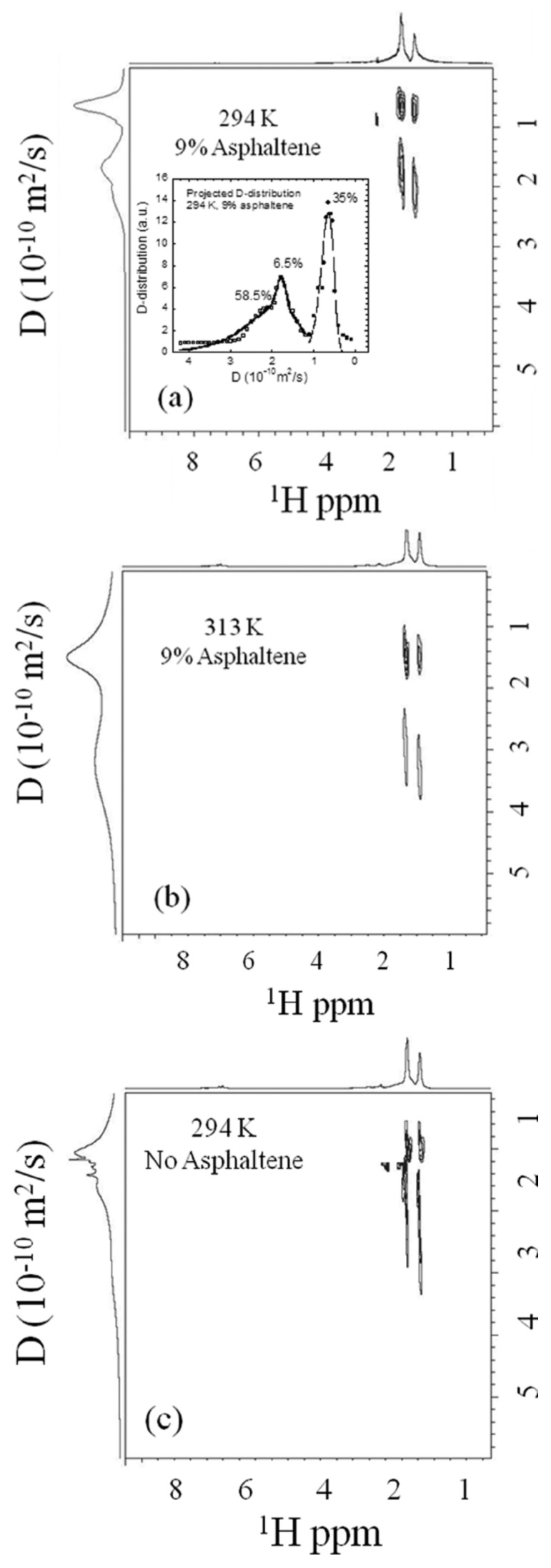
2D DOSY NMR spectrum correlating the ^1^H NMR (300 MHz) with translational diffusion coefficients using pulsed field gradient spin echo with bipolar gradients. Results for sample 1 at 294 K (**a**) and 313 K (**b**), and sample 2 at 294 K (**c**) [36].

**Figure 10 molecules-29-04038-f010:**
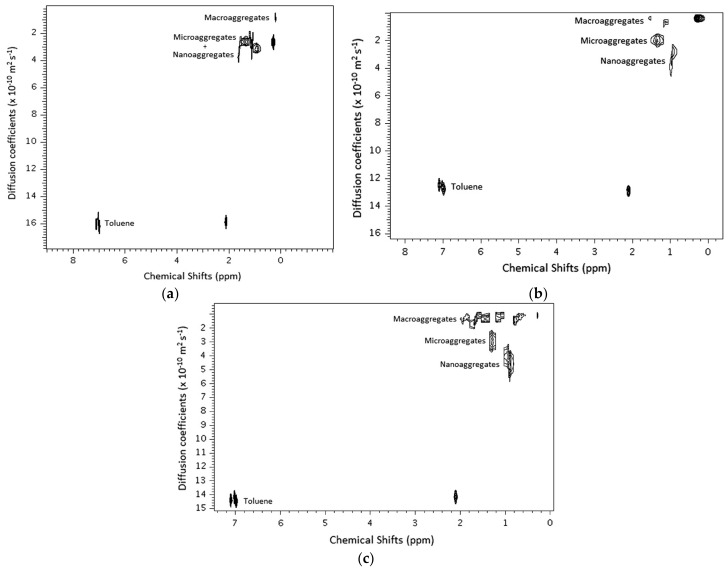
^1^H DOSY NMR spectra of asph_A (**a**), asph_B (**b**) and asph_C (**c**) (8% mass in toluene-d_8_ solution) [68].

**Figure 11 molecules-29-04038-f011:**
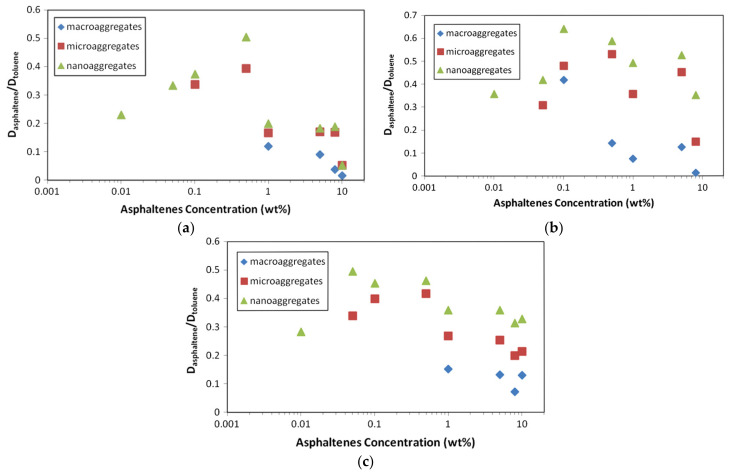
Relative diffusion of asph_A (**a**), asph_B (**b**), and asph_C (**c**) as a function of concentration. [69].

**Figure 12 molecules-29-04038-f012:**
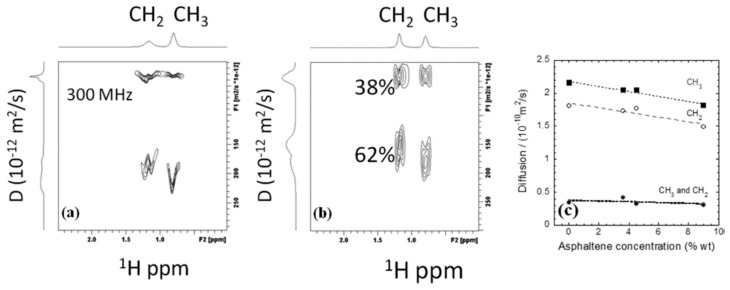
Two-dimensional DOSY NMR spectrum correlating the 1H NMR spectrum and translational diffusion coefficients D for crude oil without (**a**) and with 9% asphaltene (**b**). (**c**) Variations of D as a function of asphaltene concentration (% wt) with linear fits [69].

**Figure 13 molecules-29-04038-f013:**
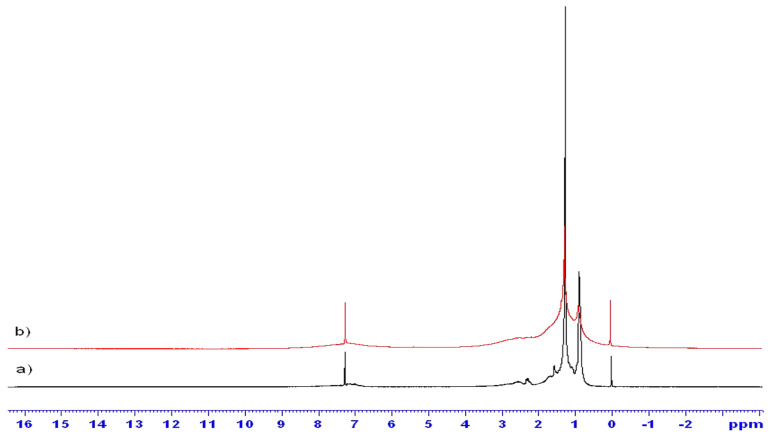
Proton spectra of (**a**) Asia 4 crude oil and (**b**) isolated asphaltene samples [70].

**Figure 14 molecules-29-04038-f014:**
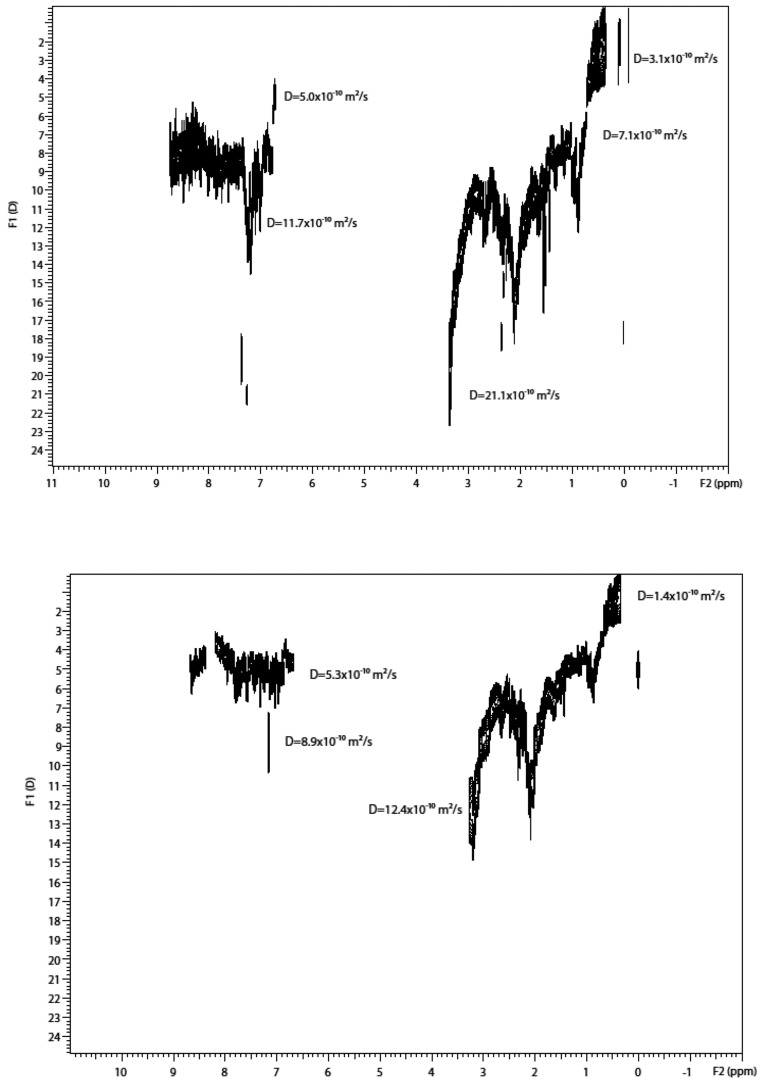
DOSY spectra of (**top**) Africa 1 crude oil and (**bottom**) Croatia 1 crude oil [70].

**Figure 15 molecules-29-04038-f015:**
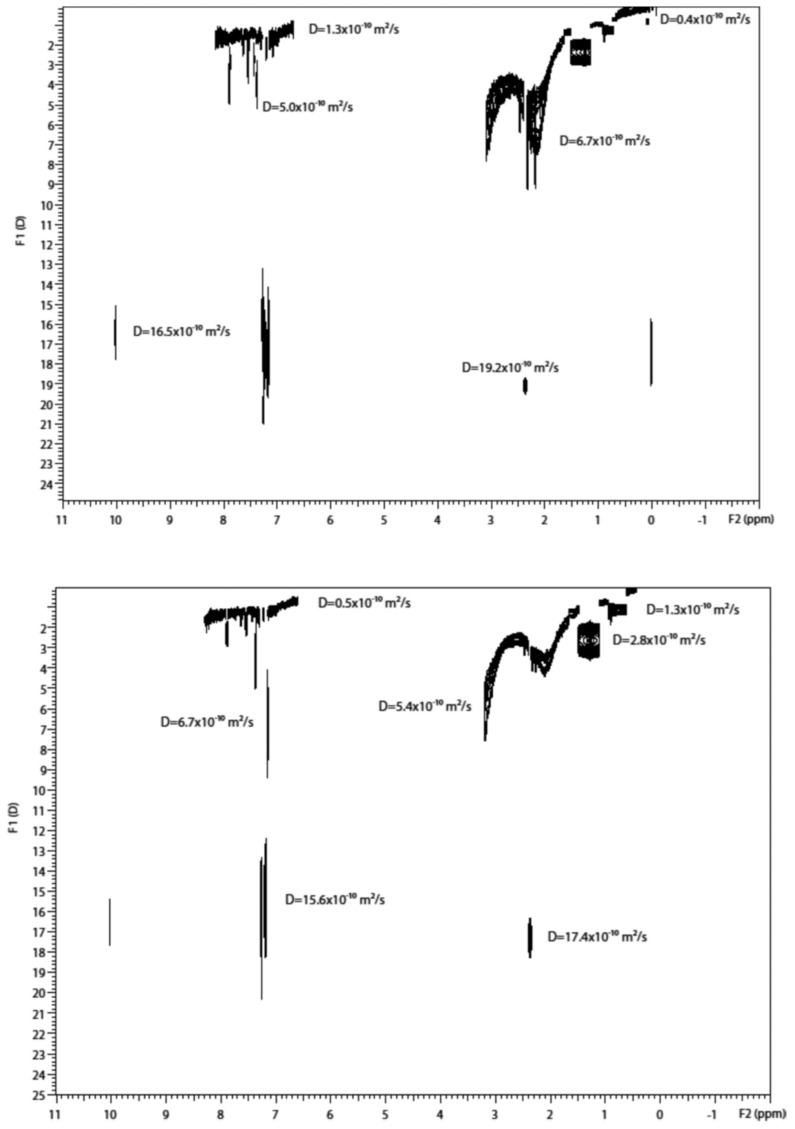
DOSY spectra of (**top**) Africa 1 crude oil and (**bottom**) Croatia asphaltene [70].

**Figure 16 molecules-29-04038-f016:**
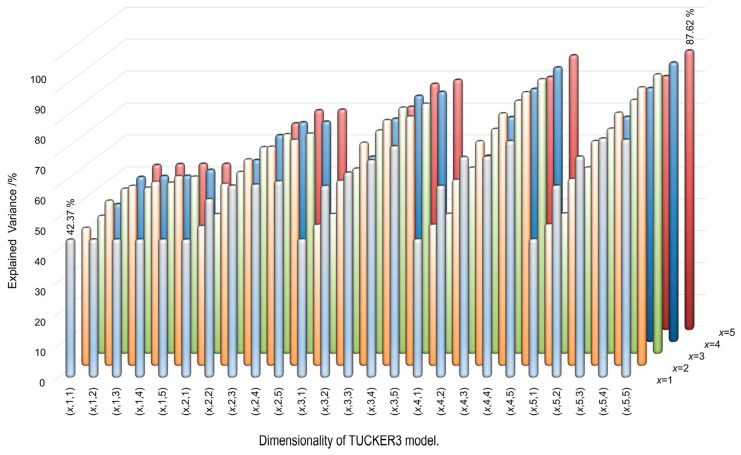
Explained variance in dependence of dimensionality for the TUCKER3 model used in the decomposition of the third-order tensor (DOSY NMR spectra) [70].

**Figure 17 molecules-29-04038-f017:**
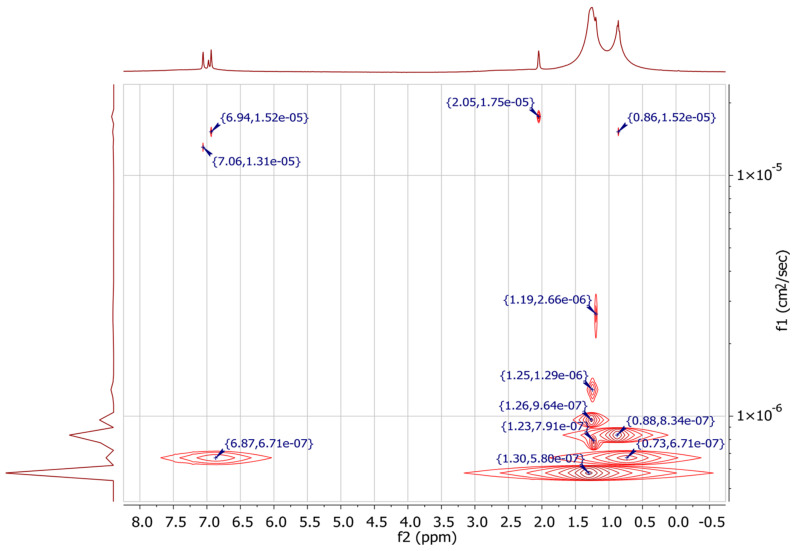
^1^H DOSY NMR spectrum of the asphaltene sample analyzed at 8 wt % in toluene-d_8_ [37].

**Table 1 molecules-29-04038-t001:** 2D NMR measurements types and applications.

Measurements	Applications	References
T_1_-T_2_ maps	Fluid-surface interaction	[72]
	Dynamics of asphaltene aggregates in crude oil	[69]
	T_1_-T_2_ correlation with viscosity η of bulk heavy crude oils	[73]
	Fluid typing	[74]
	Permeability	[75]
	Wettability	[76,77]
	Heavy oil reservoir evaluation	[78]
T_2_-Diffusion	Surface relaxivity (S/V ratio)	[79]
	Wettability	[80,81,82]
	Fluid saturation	[83]
T_2_-store-T_2_	Pore coupling	[84]
	Asphaltene deposition	[85]
	Diffusion exchange	[86]

## Data Availability

Not applicable.

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
