# Peer review of "The Use of Nuclear Magnetic Resonance Spectroscopy (NMR) to Characterize Bitumen Used in the Road Pavements Industry: A Review"

_molecules, 2024, doi:10.3390/molecules29174038_

Round 1

Reviewer 1 Report

Comments and Suggestions for Authors
The review aims to the progress about the use of NMR on bitumen used in road pavements industry. Diverse applications in determining bitumen content, group composition, molecular dynamics, and 21 interaction with additives has been discussed, and various NMR techniques have been analyzed. But some problems need to be solved.
1.  In the introduction, background on bitumen used in road pavements have not been discussed in detailed. The meaningful is lacking, and which is needed to re-organized. 2. The compariation discussion among different types of NMR is lacking, and they should be analyzed and the results should be given.
3. For all the images, they are nonstandard and should be revised.
Comments on the Quality of English Language
The review aims to the progress about the use of NMR on bitumen used in road pavements industry. Diverse applications in determining bitumen content, group composition, molecular dynamics, and 21 interaction with additives has been discussed, and various NMR techniques have been analyzed. But some problems need to be solved.
1.  In the introduction, background on bitumen used in road pavements have not been discussed in detailed. The meaningful is lacking, and which is needed to re-organized. 2. The compariation discussion among different types of NMR is lacking, and they should be analyzed and the results should be given.
3. For all the images, they are nonstandard and should be revised.

Author Response

Referee 1

The review aims to the progress about the use of NMR on bitumen used in road pavements industry. Diverse applications in determining bitumen content, group composition, molecular dynamics, and 21 interaction with additives has been discussed, and various NMR techniques have been analyzed. But some problems need to be solved.

  1. In the introduction, background on bitumen used in road pavements have not been discussed in detailed. The meaningful is lacking, and which is needed to re-organized.

We revised the introduction adding some information on the bitumen used for road pavements

  1. The compariation discussion among different types of NMR is lacking, and they should be analyzed and the results should be given.

In the section “5. Comparison between NMR techniques used in bitumen analysis”, the differences of the various NMR techniques are illustrated and compared depending on the type of information, structural or dynamic, that one intends to investigate in bitumen research.

  1. For all the images, they are nonstandard and should be revised.

The images are from literature and we are asking the permission to use in the review.

Reviewer 2 Report

Comments and Suggestions for Authors

The manuscript presents a summary of knowledge about Nuclear magnetic spectroscopy for characterizing materials and its use for analysis of the composition of bitumens from crude oils. The scope of the manuscript is suitable for publication in this journal. The authors made a good work in this field.

Some comments and corrections are necessary for the publication of the manuscript:

Revise references to all figures in the text (e.g. line 78, line 443, line 455).

In Chapter 1.2, lines 126-130 are repeated in Chapter 1.1.

I disagree with the authors' statement about missing structure/property correlations "But a rational structure/property correlation is unfortunately missing in this case." The test procedure of bitumen empirical properties has been known for over 100 years and such analyses have been done.

Chapter 2.2 describes performing a relaxometric analysis, but the purpose of such an analysis is missing.

A few formal comments:

Revise references to the literature for accuracy and standardize format (line 129 “1113”, line 158 “25”, line 183 “[26]”, line 196 “[30]”).

Why does the numbering of the figures start from zero.

Check the formatting of the equations.

Figure 0, to add the x-axis designation.

Figure 0b, the difference between the sample with 9% and 0% asphaltene content is not clear. Need to be corrected.

The figure description should include information on what the figure shows. More detailed information is given in the text. This information in manuscript is repeated (in figures and in text). Check in all figures.

Check the format of the figures (e.g. axis descriptions).

Author Response

Referee 2

The manuscript "The Use of Nuclear Magnetic Resonance Spectroscopy (NMR) to Characterize Bitumen Used in Road Pavements Industry: A Review" is a well-researched and informative review that significantly contributes to the understanding of NMR applications in bitumen analysis. With some revisions to improve readability, clarity, and engagement, it has the potential to be a highly valuable resource for researchers and professionals in the field of road pavements and materials science:

-The manuscript is heavily laden with technical jargon, which might be challenging for readers not deeply familiar with NMR spectroscopy or bitumen chemistry. Including more simplified explanations could help broaden the audience.

We are aware that NMR techniques can be somewhat cumbersome and hard to understand for non-experts in this type of spectroscopy. On the other hand, we are also aware of the risk of transmitting incomplete information in an attempt to simplify basic NMR concepts such as, for example, the different pulse sequences that characterize and differentiate the most widespread NMR methods. We are certain that the various explanatory reviews cited overall in the text can contribute to making NMR spectroscopy more accessible and understandable for researchers and professionals involved in various fields of investigation in the science of complex materials.

-Although the manuscript provides a comprehensive review, some sections, particularly those discussing advanced NMR techniques, could benefit from additional technical details. This would help readers who are less familiar with NMR to better understand the methodologies. Recent research on 2D NMR Relaxation Relayed Correlation Spectroscopy methods for polymers and bitumen has shown significant advancements in the precision and efficiency of physical-chemestry characterization [ 10.3390/polym14235332 10.1016/j.fuel.2018.02.059].

According to the referee’s suggestion we added, at the end of paragraph 7 “future directions” section, some considerations about recent research exploiting one of the articles suggested by the referee (the other one deals with supercritical CO2 sorption on polymers, probably the DOI was mistyped by the referee…). We thank the referee for suggesting us the reference: although it deals with siliceous systems, the wettability dynamics of asphaltenes onto them can give interesting ideas about the physico-chemical behavior of asphaltenic molecules. However, we were cautious in giving further technical details of NMR techniques, in favor to a clearer focus on the scientific problematics and examples of applications. This for a more easy-to-understand text to keep the audience as broad as possible. Coherently, in fact, this is what the referee him/herself suggested in the previous comment. We therefore reminded the reader to the cited literature for more technical details.

-The case studies are informative but could be more structured. Grouping them into sub-sections based on the type of NMR technique used or the specific aspect of bitumen characterization studied would improve organization.

We organized the paper according to the referee suggestions

-The conclusion summarizes the key points well but could be more impactful by highlighting the most significant contributions of NMR spectroscopy to bitumen research and its future potential.

According to the referee’s suggestion, the conclusion section was entirely rewritten, highlighting all the applications of NMR in bituminous materials, the future trends, the difficulties in understand and using NMR by unskilled researchers and how we tried to face such difficulties to make NMR spectroscopy more accessible and understandable to researchers and professionals involved in various fields of investigation of bitumens and related materials.

Reviewer 3 Report

Comments and Suggestions for Authors

The manuscript "The Use of Nuclear Magnetic Resonance Spectroscopy (NMR) to Characterize Bitumen Used in Road Pavements Industry: A Review" is a well-researched and informative review that significantly contributes to the understanding of NMR applications in bitumen analysis. With some revisions to improve readability, clarity, and engagement, it has the potential to be a highly valuable resource for researchers and professionals in the field of road pavements and materials science:

-The manuscript is heavily laden with technical jargon, which might be challenging for readers not deeply familiar with NMR spectroscopy or bitumen chemistry. Including more simplified explanations could help broaden the audience.

-Although the manuscript provides a comprehensive review, some sections, particularly those discussing advanced NMR techniques, could benefit from additional technical details. This would help readers who are less familiar with NMR to better understand the methodologies. Recent research on 2D NMR Relaxation Relayed Correlation Spectroscopy methods for polymers and bitumen has shown significant advancements in the precision and efficiency of physical-chemestry characterization [ 10.3390/polym14235332 10.1016/j.fuel.2018.02.059]. 

-The case studies are informative but could be more structured. Grouping them into sub-sections based on the type of NMR technique used or the specific aspect of bitumen characterization studied would improve organization.

-The conclusion summarizes the key points well but could be more impactful by highlighting the most significant contributions of NMR spectroscopy to bitumen research and its future potential.

Author Response

Referee 3

The manuscript presents a summary of knowledge about Nuclear magnetic spectroscopy for characterizing materials and its use for analysis of the composition of bitumens from crude oils. The scope of the manuscript is suitable for publication in this journal. The authors made a good work in this field.

Some comments and corrections are necessary for the publication of the manuscript:

Revise references to all figures in the text (e.g. line 78, line 443, line 455).

We did

In Chapter 1.2, lines 126-130 are repeated in Chapter 1.1

I disagree with the authors' statement about missing structure/property correlations "But a rational structure/property correlation is unfortunately missing in this case." The test procedure of bitumen empirical properties has been known for over 100 years and such analyses have been done.

We agree with the referee’s comment. We just wanted to express the idea that the structure/property correlations is always a hard task in such complex materials like bitumens and cannot give simple relationship like in simple systems. We apologize for our clumsy expression. We changed the text accordingly.

Chapter 2.2 describes performing a relaxometric analysis, but the purpose of such an analysis is missing.

 We added the ILT to the NMR relaxation, this methodology is used to analyze the bitumen

A few formal comments:

Revise references to the literature for accuracy and standardize format (line 129 “1113”, line 158 “25”, line 183 “[26]”, line 196 “[30]”).

Why does the numbering of the figures start from zero.

Check the formatting of the equations.

Figure 0, to add the x-axis designation.

Figure 0b, the difference between the sample with 9% and 0% asphaltene content is not clear. Need to be corrected.

The figure description should include information on what the figure shows. More detailed information is given in the text. This information in manuscript is repeated (in figures and in text). Check in all figures.

Check the format of the figures (e.g. axis descriptions).

We did